# A phylogenetically-restricted essential cell cycle progression factor in the human pathogen *Candida albicans*

Priya Jaitly[1], Mélanie Legrand[2], Abhijit Das [1,5], Tejas Patel [1,5], Murielle Chauvel[2], Corinne Maufrais [3], Christophe d'Enfert [2✉] & Kaustuv Sanyal [1,4✉]

Chromosomal instability caused by cell division errors is associated with antifungal drug resistance in fungal pathogens. Here, we identify potential mechanisms underlying such instability by conducting an overexpression screen monitoring chromosomal stability in the human fungal pathogen *Candida albicans*. Analysis of ~1000 genes uncovers six chromosomal stability (*CSA*) genes, five of which are related to cell division genes of other organisms. The sixth gene, *CSA6*, appears to be present only in species belonging to the CUG-Ser clade, which includes *C. albicans* and other human fungal pathogens. The protein encoded by *CSA6* localizes to the spindle pole bodies, is required for exit from mitosis, and induces a checkpoint-dependent metaphase arrest upon overexpression. Thus, Csa6 is an essential cell cycle progression factor that is restricted to the CUG-Ser fungal clade, and could therefore be explored as a potential antifungal target.

[1] Molecular Mycology Laboratory, Molecular Biology and Genetics Unit, Jawaharlal Nehru Centre for Advanced Scientific Research, Bangalore, India. [2] Institut Pasteur, Université Paris Cité, INRAE, USC2019, Unité Biologie et Pathogénicité Fongiques, F-75015 Paris, France. [3] Institut Pasteur, Université Paris Cité, Bioinformatics and Biostatistics Hub, F-75015 Paris, France. [4] Osaka University, Suita, Osaka, Japan. [5] These authors contributed equally: Abhijit Das, Tejas Patel. ✉email: christophe.denfert@pasteur.fr; sanyal@jncasr.ac.in

The primary objective of cell division is to ensure genome stability by preserving and transferring the genetic material with high fidelity into progeny. Genome stability is achieved by proper execution of key cell cycle events, including equal segregation of the duplicated chromosomes during the M phase. The integrity and fidelity of these cell cycle events in turn are monitored by various cell cycle checkpoints. Failure of any of the error-correcting mechanisms at cell cycle checkpoints can introduce genetic alterations, causing genomic instability in progeny. Genome instability can occur as a consequence of either point mutations, insertions and deletions of bases in specific genes or gain, loss and rearrangements of chromosomes[1]. The latter is also referred to as chromosome instability (CIN)[1]. CIN has been intimately associated with the generation of aneuploidy[2] and is one of the potential drivers of human genetic and neuro-degenerative disorders[3,4], aging[5] and several cancers[6]. While considered harmful for a cell or an organism, CIN may also contribute to generating variations and help in driving evolution, especially in unicellular and primarily asexual eukaryotes[7,8].

The current understanding of the mechanisms underlying genome stability has evolved through studies in a range of biological systems from unicellular yeasts to more complex metazoa, including humans. These studies highlighted concerted actions of genes involved in (a) high-fidelity DNA replication and DNA damage repair, (b) efficient segregation of chromosomes and (c) error-correcting cellular surveillance machinery[9,10] in the maintenance and inheritance of a stable genome. In recent years, large-scale screenings of loss-of-function[11], reduction-of-function[12] and overexpression[13–16] mutant collections in the budding yeast *Saccharomyces cerevisiae* have appended the list of genome stability-regulators by identifying uncharacterized proteins as well as known proteins with functions in other cellular processes. However, considering the vast diversity of the chromosomal segregation mechanisms in eukaryotes, it is conceivable that many genes involved in genome maintenance are yet to be discovered. Hence, additional genetic screens in a wide range of organisms will facilitate the identification of previously uncharacterized regulators of genome stability.

The ascomycetous yeast *Candida albicans* is emerging as an attractive unicellular model for studying eukaryotic genome biology[17]. *C. albicans*, a commensal of both the gastrointestinal and genital tracts, remains the most frequently isolated fungal species worldwide from the patients diagnosed with candidemia or other nosocomial *Candida* infections[18,19]. The diploid genome of *C. albicans* shows remarkable plasticity in terms of ploidy, single nucleotide polymorphism (SNP), loss of heterozygosity (LOH), copy number variations, and CIN events[17,20]. Although LOH can be detected on all the chromosomes of *C. albicans*, the presence of recessive lethal or deleterious alleles on some haplotypes prevents one of the haplotypes or even a part of it from existing in the homozygous state[17]. This homozygous bias has been observed for chromosomes R (ChR), 2 (Ch2), 4 (Ch4), 6 (Ch6) and 7 (Ch7)[21,22]. LOH and aneuploidy-driven CIN has serious phenotypic consequences in *C. albicans* such as conferring resistance to antifungals[23–28] or helping *C. albicans* adapt to different host niches[29–31]. With increasing instances of *Candida* infections and emerging antifungal resistance, there is an immediate need to identify additional fungus-specific molecular targets that may aid the development of antifungal therapies. In addition, the remarkable ability of *C. albicans* to tolerate CIN in the form of whole chromosome loss, isochromosome formation, chromosome truncation, or mitotic crossing-over[17,20,32] raises intriguing questions on the functioning of genome stability-regulators in this fungal pathogen.

Of utmost importance to maintain genomic integrity is the efficient and error-free segregation of the replicated chromosomes.

In most eukaryotes including *C. albicans*, the assembly of a macromolecular protein complex, called the kinetochore (KT), on CENP-A (Cse4 in budding yeasts) containing centromeric chromatin mediates chromosome segregation during mitosis[33–36]. The KT acts as a bridge between a chromosome and the connecting microtubules (MTs), emanating from the spindle pole bodies (SPBs), the functional homolog of centrosomes in mammals[37]. The subsequent attachment of sister KTs to opposite spindle poles then promotes the formation of a bipolar mitotic spindle that drives the separation of the duplicated chromosomes during anaphase[38], after which cells exit mitosis and undergo cytokinesis[39–41]. In *C. albicans*, KT proteins remain clustered throughout the cell cycle and are shown to be essential for viability and mitotic progression[33,36,42,43]. In addition, genes involved in homologous recombination, such as *MRE11* and *RAD50*, and DNA damage checkpoint pathways, including *MEC1*, *RAD53* and *DUN1*, are required to prevent genome instability in *C. albicans*[44–46]. A recent screen, using a collection of 124 over-expression strains, has identified three additional genes, namely, *CDC20*, *BIM1*, and *RAD51*, with a role in genome maintenance as indicated by increased LOH-driven CIN upon overexpression in *C. albicans*[47]. Currently, only a minor fraction of the *C. albicans* gene armamentarium has been evaluated for their roles in genome stability. Systematic approaches are thus needed to exhaustively define the drivers of *C. albicans* genome maintenance and outline species-specific processes as well as commonalities with other eukaryotes.

Here, we describe a large-scale screen aimed at identifying regulators of genome stability in a clinically relevant fungal model system. Our screen, involving ~20% of the *C. albicans* ORFeome, has identified Csa6, a yet unknown player of genome stability, as a critical regulator of cell cycle progression in *C. albicans*. Overall, this is the first-ever report of such a screen at this scale in *C. albicans* and provides a framework for identifying regulators of eukaryotic genome stability, some of which may serve as potential targets for therapeutic interventions of fungal infections.

## Results

**A reporter system for monitoring chromosome stability in *C. albicans*.** To understand the molecular mechanisms underlying genome stability in a fungal pathogen, we developed a reporter system in *C. albicans* in which whole chromosome loss can be distinguished from other events such as break-induced replication, gene conversion, chromosome truncation or mitotic crossing over[22,47]. In our prior work, a loss-of-heterozygosity (LOH) reporter strain was developed for use in *C. albicans*[22]. In the LOH reporter system, GFP and BFP genes, linked to *ARG4* and *HIS1* genes, respectively, have been integrated at the identical intergenic locus on the left arm of both homologs of chromosome 4, Ch4A and Ch4B, respectively (Fig. 1a, Supplementary Fig. 1a)[47]. Consequently, cells express both GFP and BFP as analyzed by flow cytometry and are prototrophic for *ARG4* and *HIS1* genes, unless a chromosome instability (CIN) event causes loss of one of the two loci (Fig. 1a, b)[22]. To differentiate whole chromosome loss from other events that may lead to loss of one of the two reporter loci, we modified the LOH reporter strain by integrating a red fluorescent protein (RFP) reporter gene, associated with the hygromycin B (hyg B) resistance marker, on the right arm of Ch4B (Fig. 1a, Supplementary Fig. 1a). The RFP reporter insertion is sufficiently distant from the BFP locus that loss of both BFP and RFP signal (and of their linked auxotrophic/resistance markers) is indicative of loss of Ch4B, rather than a localized event causing loss of the *BFP-HIS1* reporter insertion (Fig. 1a, Supplementary Fig. 1a). Notably, while loss of Ch4A cannot be tolerated due to the presence of recessive lethal alleles on Ch4B[22], loss of Ch4B leads to the formation of small colonies that grow

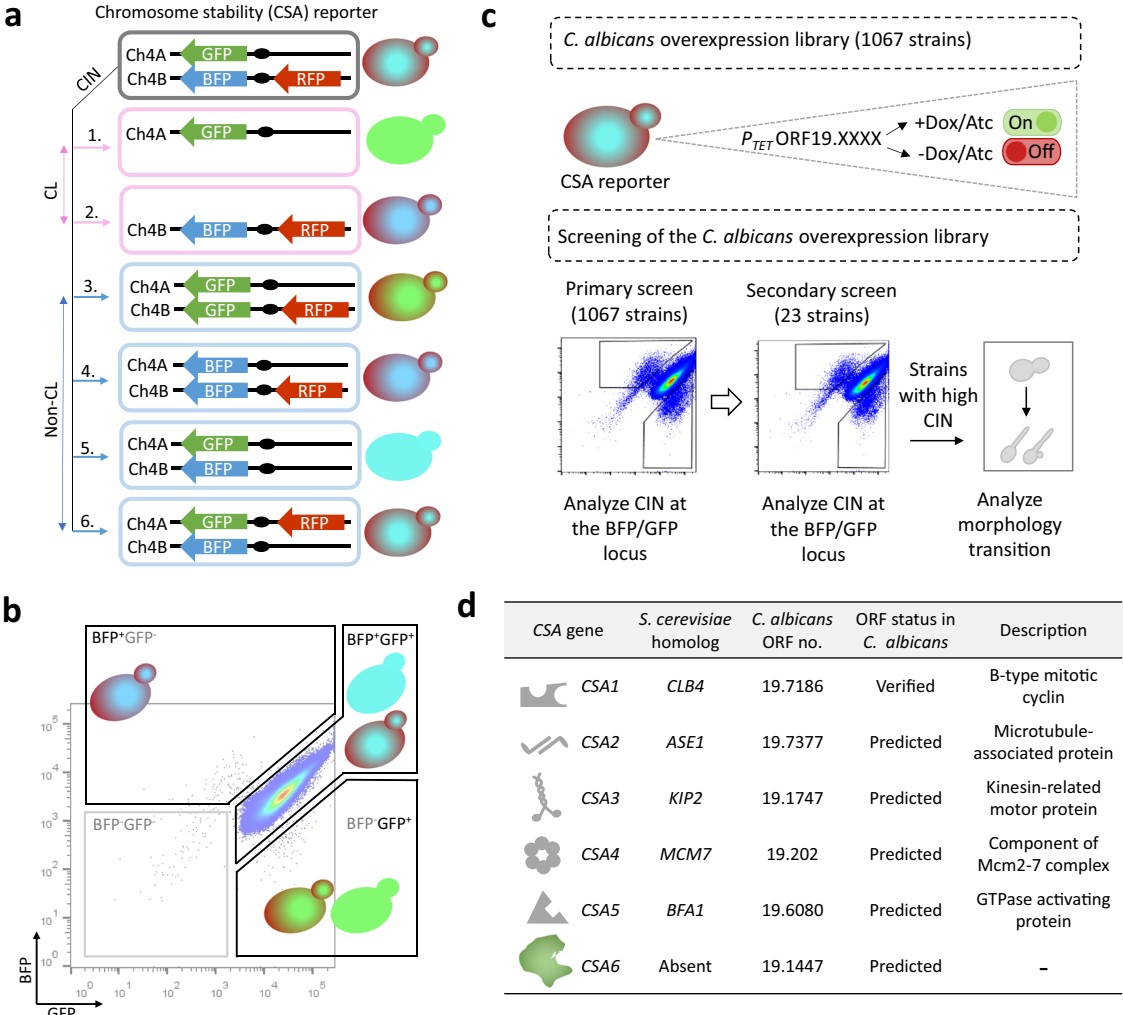

**Fig. 1 A medium-throughput protein overexpression screen identifies a set of *CSA* genes in *C. albicans*. a** Possible outcomes of CIN at the BFP/GFP and RFP loci. *1-4*, CIN at the BFP or GFP locus, because of either chromosome loss (CL) or non-CL events such as break-induced replication, gene conversion, chromosome truncation or mitotic crossing over, will lead to the expression of either GFP or BFP expressing genes. CIN due to CL can be specifically identified by the concomitant loss of BFP and RFP, as shown in *1*. *5 and 6*, cells undergoing non-CL events at the RFP locus will continue to express BFP and GFP. **b** Flow cytometric analysis of the BFP/GFP density profile of empty vector (EV) (CaPJ150) containing BFP, GFP and RFP genes. The majority of the cells are positive for both BFP and GFP (BFP⁺GFP⁺). A minor fraction of the population had lost either one of the markers (BFP⁺GFP⁻ or BFP⁻GFP⁺) or both the markers (BFP⁻GFP⁻), indicating spontaneous instability of this locus[47]. Approximately 1 million events are displayed. **c** Pictorial representation of the screening strategy employed for identifying *CSA* genes in *C. albicans*. Briefly, a library of *C. albicans* overexpression strains (1067), each carrying a unique ORF under the tetracycline-inducible promoter, $P_{TET}$, was generated using the CSA reporter (CEC5201) as the parent strain. The library was then analyzed by primary and secondary screening methods to identify *CSA* genes. In the primary screen, CIN frequency at the BFP/GFP locus in the individual 1067 overexpression strains was determined using flow cytometry. Overexpression strains exhibiting increased CIN (23 out of 1067) were taken forward for secondary screening. The secondary screen involved revalidation of the primary hits for increased CIN at the BFP/GFP locus by flow cytometry. Strains that reproduced the increased CIN phenotype were further examined for yeast to filamentous transition by microscopy. **d** A brief overview of the *CSA* genes identified from the overexpression screen (6 out of 1067). Functional annotation of genes is based on the information available either in *Candida Genome Database* (www.candidagenome.org) or *Saccharomyces Genome Database* (www.yeastgenome.org) on August 1, 2021.

into larger colonies following duplication of Ch4A[47]. Thus, the absence of both *BFP-HIS1* and *RFP-HYG B* but continued presence of *GFP-ARG4* in the modified reporter strain, which we named as chromosome stability (CSA) reporter, enables us to monitor the loss of Ch4B in a population. The fluorescence intensity profile of GFP, BFP and RFP in the CSA reporter was validated by flow cytometry (Supplementary Fig. 1b). To functionally validate the CSA reporter system, we overexpressed *CDC20*, a gene important for anaphase onset, activation of spindle assembly checkpoint and whose overexpression is known to cause whole chromosome loss in *C. albicans*[47]. We analyzed the BFP/GFP density plots in various control strains

(Supplementary Fig. 1c) and monitored the loss of BFP/GFP signal in cells overexpressing *CDC20* (*CDC20^OE^*) by flow cytometry. As reported earlier[47], the *CDC20^OE^* strain displayed a higher population of BFP⁺GFP⁻ and BFP⁻GFP⁺ cells as compared to the empty vector (EV) control indicating increased CIN in the *CDC20^OE^* mutant (Supplementary Fig. 1d, e). Next, we isolated BFP⁻GFP⁺ cells of EV and *CDC20^OE^* using flow cytometry and plated them for subsequent analysis of auxotrophic/resistance markers (Supplementary Fig. 1f). As noted above, upon incubation of the sorted BFP⁻GFP⁺ cells, we observed the appearance of both small and large colonies (Supplementary Fig. 1f). Small colonies have been previously shown to be the

result of loss of Ch4B homolog and are predicted to be a consequence of Ch4A monosomy, eventually yielding large colonies upon reduplication of Ch4A[47]. We, therefore, performed the marker analysis on large colonies and found that 85% of the BFP⁻GFP⁺ derived colonies of $CDC20^{OE}$ mutant concomitantly lost both *HIS1* and *HYG B* but retained *ARG4* (Supplementary Fig. 1g) suggesting the loss of Ch4B homolog; flow cytometry analysis further confirmed the loss of BFP and RFP signals in these colonies. The remaining 15% of colonies retained *GFP-ARG4* and *RFP-HYG B* but not *BFP-HIS1* (Supplementary Fig. 1g) indicating that more localized events including gene conversion, rather than whole chromosome loss, were responsible for the loss of the BFP signals in these cells. The above data indicate that the CSA reporter system that we engineered enables precise monitoring of the whole chromosome loss event in a population and enables large-scale screening of this phenotype.

**Medium-throughput screening of *C. albicans* overexpression strains identifies regulators of genome stability.** Systematic gene overexpression is an attractive approach for performing large-scale functional genomic analysis in *C. albicans*, a diploid ascomycete. Using a recently developed collection of *C. albicans* inducible overexpression plasmids (M.C., S. Bachellier-Bassi, A.M. Guérout, K.K. Lee, C. Maufrais, E. Permal, J. Pipoli Da Fonseca, S. Znaidi, D. Mazel, C.A. Munro, C.d'E., M.L., unpublished results) and the CSA reporter strain described above, we generated a library of 1067 *C. albicans* inducible overexpression strains. Each of these strains, carrying a unique ORF under the control of the $P_{TET}$ promoter, could be induced for overexpression after anhydrotetracycline (Atc) or doxycycline (Dox) addition (Fig. 1c)[47,48]. To identify regulators of genome stability, we carried out a primary screen with these 1067 overexpression strains by individually analyzing them for the loss of BFP/GFP signals by flow cytometry (Fig. 1c, Supplementary Fig. 2a, Supplementary Data 1). Our primary screening identified 23 candidate genes (out of 1067) whose overexpression resulted in ≥2-fold increase in the BFP⁺GFP⁻ and BFP⁻GFP⁺ population relative to the EV (Supplementary Table 1, 2). Next, we carried out a secondary screen with these 23 overexpression strains to revalidate the loss of BFP/GFP markers by flow cytometry (Fig. 1c, Supplementary Fig. 2b). As genotoxic stress is intimately linked with polarized growth in *C. albicans*[17,49], we microscopically examined the overexpression strains exhibiting higher instability at the BFP/GFP locus during secondary screening for any morphological transition (Fig. 1c, Supplementary Fig. 2b). While overexpression of 17 genes (out of 23) could not reproduce the BFP/GFP loss phenotype, overexpression of the six genes resulted in ≥2-fold increase in the BFP⁺GFP⁻ or BFP⁻GFP⁺ population as compared to the EV, with three genes (out of 6) inducing polarized growth upon overexpression (Supplementary Fig. 3a, b). These six genes, which we referred to as CSA genes, include *CSA1 (CLB4)*, *CSA2 (ASE1)*, *CSA3 (KIP2)*, *CSA4 (MCM7)*, *CSA5 (BFA1)* and *CSA6* coded by *ORF19.1447* of unknown function (Fig. 1d).

**Molecular mechanisms underlying CIN in *CSA* overexpression mutants.** Out of the six CSA genes, overexpression of three genes, namely, $CSA1^{CLB4}$, $CSA2^{ASE1}$ and $CSA3^{KIP2}$ caused little or no change in the morphology of *C. albicans* (Supplementary Fig. 3a) but triggered CIN at the BFP/GFP locus, indicated by an expansion of the BFP⁺GFP⁻ and BFP⁻GFP⁺ population in the flow cytometry density plots (Supplementary Fig. 3b, c). To further dissect the molecular mechanisms leading to the loss of BFP/GFP signals in these mutants, we sorted BFP⁻GFP⁺ cells of these mutants and plated them for *GFP-ARG4*, *BFP-HIS1* and

*RFP-HYG B* analysis, as described previously for the $CDC20^{OE}$ mutant. We observed that a majority of the large BFP⁻GFP⁺ derived colonies of $CSA1^{CLB4}$, $CSA2^{ASE1}$ and $CSA3^{KIP2}$ overexpression mutants lost *BFP-HIS1* but retained *RFP-HYG B* and *GFP-ARG4* (Supplementary Fig. 3d), suggesting that localized genome instability events, rather than whole chromosome loss events, contributed to the high percentage of BFP⁻GFP⁺ cells in these mutants. In contrast, most small colonies (>95%) of each $CSA1^{CLB4}$, $CSA2^{ASE1}$ and $CSA3^{KIP2}$ overexpression mutants lost both *HIS1* and *HYG B* while retaining *ARG4*, indicating small colonies are monosomic for Ch4A (Supplementary Fig. 3e). This is consistent with our previous work[47].

Overexpression of each of the remaining three genes, namely $CSA4^{MCM7}$, $CSA5^{BFA1}$ or *CSA6*, drastically altered the morphology of *C. albicans* cells by inducing polarized/filamentous growth (Supplementary Fig. 3a). A connection between morphological switches and genotoxic stresses has been established in the polymorphic fungus *C. albicans*, wherein polarized growth is triggered in response to improper cell cycle regulation[42,43,49–51]. Flow cytometric analysis of cell cycle progression revealed that overexpression of $CSA4^{MCM7}$, $CSA5^{BFA1}$ and *CSA6* shifted cells towards the 4N DNA content (Supplementary Fig. 3f). To further determine the cell cycle phase associated with the 4N shift, we compared nuclear segregation patterns (Hoechst staining for DNA and CENP-A/Cse4 localization for KT) and spindle dynamics (separation of Tub4 foci) in these overexpression mutants with those of the EV control (Supplementary Fig. 3g). Our results suggested the 4N shift in $CSA4^{MCM7}$ and *CSA6* overexpression mutants was a result of G2/M arrest, indicated by a high percentage of large-budded cells with unsegregated DNA mass and improperly separated SPBs (Supplementary Fig. 3g). In contrast, the 4N shift upon $CSA5^{BFA1}$ overexpression was a consequence of late anaphase/telophase arrest, shown by an increased number of large-budded cells with segregated nuclei and SPBs (Supplementary Fig. 3g). Taken together, our results indicate that the polarized growth in $CSA4^{MCM7}$, $CSA5^{BFA1}$ and *CSA6* overexpression mutants is a probable outcome of improper cell cycle progression.

Two CSA genes, namely $CSA2^{ASE1}$ and $CSA5^{BFA1}$, gave rise to similar overexpression phenotypes in both *S. cerevisiae* and *C. albicans* (Table 1). While phenotypes related to $CSA4^{MCM7}$ and *CSA6* overexpression in *S. cerevisiae* or other related organisms remained unreported, the overexpression phenotypes of the remaining CSA genes were along the lines of their roles in cell cycle functioning, as reported in *S. cerevisiae* (Table 1, Fig. 1d). Altogether, our results validated the role of CSA genes in regulating genome stability in *C. albicans*. While overexpression of either $CSA1^{CLB4}$, $CSA2^{ASE1}$ or $CSA3^{KIP2}$ induced CIN mostly through non-chromosomal loss events, the effect of overexpression of either $CSA4^{MCM7}$, $CSA5^{BFA1}$ or *CSA6* was so drastic that the *C. albicans* mutants were arrested at different cell cycle phases with G2/M equivalent DNA content (4N) and thus were unable to complete the mitotic cell cycle.

To further assess the extent of genome changes associated with overexpression of *CSA6*–a gene of unknown function, we sequenced the genome of 4 independent large BFP⁻GFP⁺ colonies in which LOH has arisen on Ch4 either spontaneously (in YPD cultures) or as the consequence of *CSA6* overexpression (in YPD + Atc cultures) (Supplementary Fig. 4a). While we confirmed the LOH events occurring on Ch4, as selected by flow cytometry, we did not observe additional large-scale genome changes other than the LOH event we were selecting for on Ch4 (Supplementary Fig. 4b). This suggests that overexpression of *CSA6* does not change the nature of the CIN events but rather results in an increased frequency of these events in *C. albicans*.

**Table 1 Overexpression phenotypes of *CSA* genes in *C. albicans* and *S. cerevisiae*.**

| *CSA* gene | *C. albicans* ORF no. | *S. cerevisiae* homolog | Overexpression phenotype (*C. albicans*) | Overexpression phenotype (*S. cerevisiae*) | Reference |
|---|---|---|---|---|---|
| *CSA1* | 19.7186 | *CLB4* | Increased CIN involving non-CL events | Shift towards 2N (diploid) DNA content | 109 |
| *CSA2* | 19.7377 | *ASE1* | Increased CIN involving non-CL events | i) CIN involving loss of an artificial chromosome fragment or rearrangements/ gene conversion events. ii) Spindle checkpoint dependent delay in entering anaphase upon HU treatment | 14,82 |
| *CSA3* | 19.1747 | *KIP2* | Increased CIN involving non-CL events | Shift towards 2N (diploid) DNA content | 87,109 |
| *CSA4* | 19.202 | *MCM7* | Shift towards 4N (diploid) DNA content, G2/M arrest | NA | NA |
| *CSA5* | 19.608 | *BFA1* | Shift towards 4N (diploid) DNA content, anaphase arrest | Shift towards 2N (diploid) DNA content, Anaphase arrest | 110 |
| *CSA6* | 19.1447 | NA | Shift towards 4N (diploid) DNA content, G2/M arrest | NA | NA |

*NA* not available.

**Csa6 is an SPB-localizing protein, present across a subset of CUG-Ser clade fungal species**. Among the genes identified in the screen, Csa6 was the only protein without any detectable homolog in *S. cerevisiae* (Fig. 1d). This intrigued us to examine its presence across various other fungi. Phylogenetic analysis using high confidence protein homology searches and synteny-based analysis indicated that Csa6 is exclusively present in a subset of fungal species belonging to the CUG-Ser clade (Fig. 2a). Strikingly, in all these species, Csa6 was predicted to have a central coiled-coil domain (Supplementary Fig. 5a). Epitope tagging of Csa6 with a fluorescent marker (mCherry) localized it close to the KT throughout the cell cycle in *C. albicans* (Fig. 2b). In most unicellular fungi, often found proximal to the clustered KTs, are the SPB complexes[33,35,52,53]. Although neither the SPB structure nor its composition is well characterized in *C. albicans*, the majority of the SPB proteins exhibit high sequence and structural conservation from yeast to humans[54]. Hence, we re-examined Csa6 localization with two of the evolutionarily conserved SPB proteins, Tub4 and Spc110, in *C. albicans*[54,55] (Fig. 2c, d, Supplementary Fig. 5b). These results showed that Csa6 constitutively localizes to the SPBs, close to the KTs, in cycling yeast cells of *C. albicans* (Fig. 2c, d). To further analyze the proximity of Csa6 with SPB, we fluorescently labeled the homolog of one of the central plaque-associated SPB proteins of *S. cerevisiae* Cmd1 in *C. albicans*. Like *S. cerevisiae*, Cmd1 of *C. albicans* also localizes to SPBs throughout the cell cycle[56] (Fig. 2e, Supplementary Fig. 5c). In addition, we observed ring-like fluorescence signals of Cmd1 near the bud neck and cables of fluorescence in the buds[56] (Fig. 2e, Supplementary Fig. 5c). Further analysis involving localization of Csa6 with Cmd1 (Fig. 2e) confirmed that Csa6 colocalizes with multiple SPB-associated proteins including Tub4, Spc110 and Cmd1 in *C. albicans*.

**Csa6, a previously uncharacterized protein, as a key regulator of mitotic progression in *C. albicans***. While the roles of Csa6 have not been investigated before, based on our findings thus far (Supplementary Fig. 3f, g), we hypothesized that Csa6 plays an important function in cell cycle regulation and genome stability in *C. albicans*. We sought to identify the molecular pathways by which Csa6 performed its functions in *C. albicans*. For this, we again made use of the inducible $P_{TET}$ promoter system to generate a $CSA6^{OE}$ strain (CaPJ176, $P_{TET}CSA6$) in the SN148 genetic background of *C. albicans*[57] (Fig. 3a). Conditional overexpression

of TAP-tagged Csa6 (CaPJ181, $P_{TET}CSA6$-*TAP*), in presence of Atc, was confirmed by western blot analysis (Fig. 3b). The effect of $CSA6^{OE}$ (CaPJ176, $P_{TET}CSA6$) on cell cycle functioning was then investigated by flow cytometric cell cycle analysis (Fig. 3c) and microscopic examination of the nuclear division (Fig. 3d). As observed previously (Supplementary Fig. 3f, g), $CSA6^{OE}$ inhibited cell cycle progression in *C. albicans* by arresting cells in the G2/M phase, evidenced by the gradual accumulation of large-budded cells with unsegregated nuclei (Fig. 3d), possessing 4N DNA content (Fig. 3c). Some of these large-budded cells also underwent a morphological transition to an elongated bud or other complex multi-budded phenotypes (Fig. 3d), indicating cell cycle arrest-mediated morphological switching[49] due to $CSA6^{OE}$. Strikingly, continuous upregulation of Csa6 was toxic to the cells (Supplementary Fig. 6a) and led to a decrease in cell viability (Supplementary Fig. 6b) as nuclei failed to segregate in this mutant (Fig. 3d).

Nuclear segregation during mitosis is facilitated by the formation of the mitotic spindle and its dynamic interactions with chromosomes via KTs. Thus, we sought to examine both the KT integrity and the mitotic spindle morphology in the $CSA6^{OE}$ mutants. In *C. albicans*, the structural stability of the KT is a determinant of CENP-A/Cse4 stability wherein depletion of any of the essential KT proteins results in delocalization and degradation of the CENP-A/Cse4 by ubiquitin-mediated proteolysis[51]. Fluorescence microscopy and western blot analysis confirmed that Cse4 was neither delocalized (Supplementary Fig. 6c) nor degraded from centromeric chromatin (Supplementary Fig. 6d) upon $CSA6^{OE}$. Next, we analyzed the spindle integrity in $CSA6^{OE}$ mutants by tagging Tub4 (SPB) and Tub1 (MTs) with fluorescent proteins. Fluorescence microscopy analysis revealed that a large proportion (~73%) of the large-budded cells formed an unconventional rudimentary mitotic spindle structure upon $CSA6^{OE}$, wherein it had a dot-like appearance as opposed to an elongated bipolar rod-like spindle structure in EV or uninduced (-Atc) strains (Fig. 3e). This suggests that nuclear segregation defects in $CSA6^{OE}$ mutant cells are an attribute of aberrant mitotic spindle formation that might have led to the mitotic arrest.

During mitosis, surveillance mechanisms, including spindle assembly checkpoint (SAC)[58,59] and spindle positioning checkpoint (SPOC)[60,61] operate to maintain genome stability by delaying the metaphase-anaphase transition in response to improper chromosome-spindle attachments and spindle

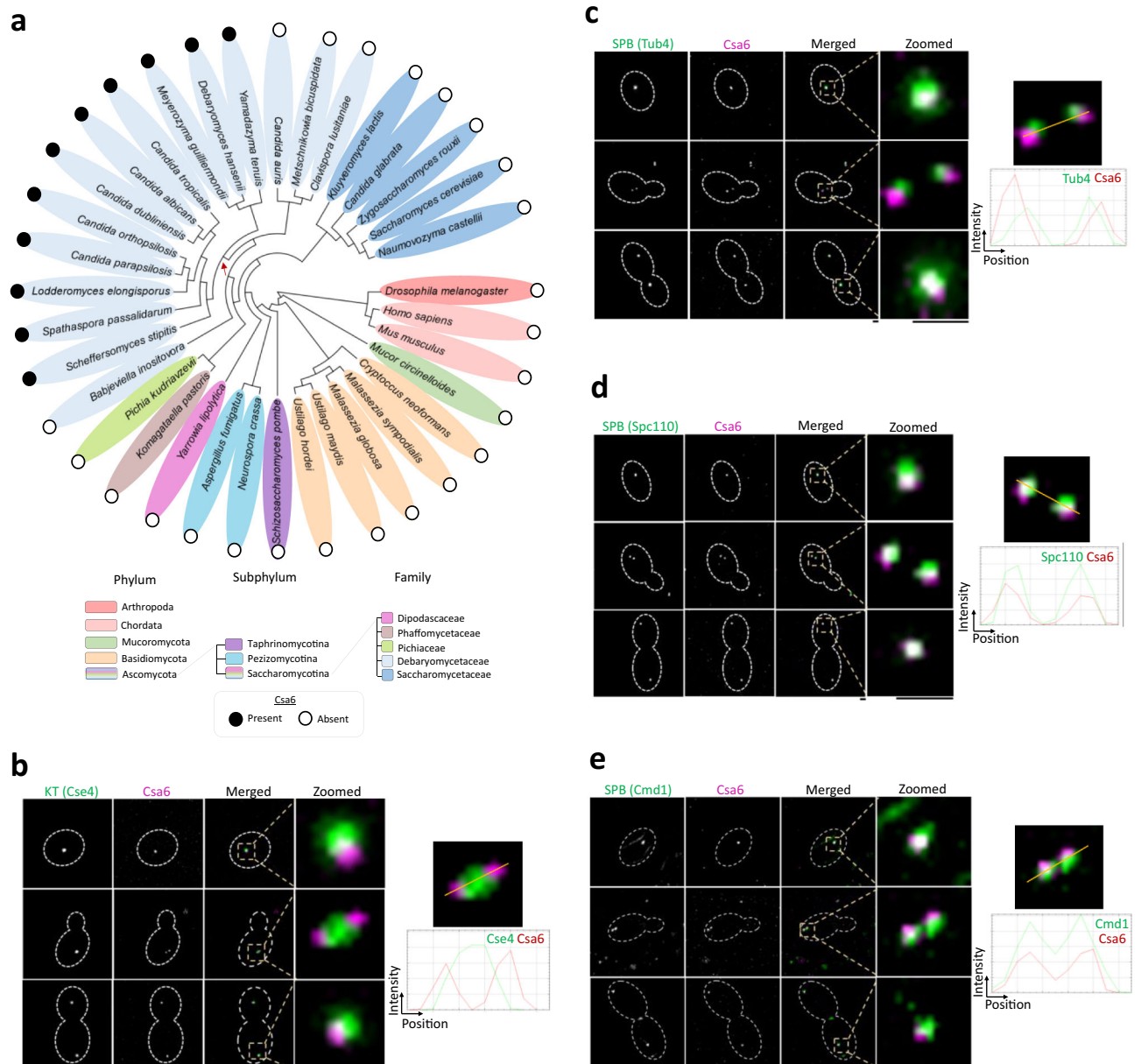

**Fig. 2 Csa6 has a selective existence across fungal phylogeny and is constitutively localized to the SPBs in *C. albicans*. a** Phylogenetic tree showing the conservation of Csa6 across the mentioned species. The presence (filled circles) or absence (empty circles) of Csa6 in every species is marked. Each taxonomic rank is color-coded. The species mentioned under the family Debaryomycetaceae belong to the CUG-Ser clade in which the CUG codon is often translated as serine instead of leucine. The red arrow points to the CUG-Ser clade lineage that acquired Csa6. Searches for Csa6 homologs (*E* value $\leq 10^{-2}$) were carried out either in the *Candida Genome Database* (www.candidagenome.org) or NCBI nonredundant protein database. **b–e** *Left*, micrographs comparing the sub-cellular localization of Csa6 with KT (Cse4) and SPB (Tub4, Spc110 and Cmd1) at various cell cycle stages. *b*, Csa6-mCherry and Cse4-GFP (CaPJ119); *c*, Csa6-mCherry and Tub4-GFP (CaPJ120); *d*, Csa6m-Cherry and Spc110-GFP (CaPJ121) and *e*, Csa6m-Cherry and Cmd1-GFP (CaPJ122). Scale bar, 1 μm. *Right*, histogram plots showing the fluorescence intensity profile of Csa6-mCherry with Cse4-GFP (**b**), Tub4-GFP (*c*), Spc110-GFP (**d**) and Cmd1-GFP (**e**) across the indicated lines. Note that Cmd1 is also localized at the bud neck and as cables inside the cell in *e*.

misorientation, respectively. We posit that the G2/M cell cycle arrest due to *CSA6^OE* in *C. albicans* could be a result of either SAC or SPOC activation. Hence, we decided to inactivate SAC and SPOC, individually, in the *CSA6^OE* strain by deleting the key spindle checkpoint genes *MAD2*[42] and *BUB2*[49], respectively. SAC inactivation in *CSA6^OE* mutant cells (Fig. 4a) led to the emergence of unbudded cells with 2N DNA content (Fig. 4b, c), indicating a bypass of the G2/M arrest caused by *CSA6^OE*. Consequently, we also observed a partial rescue of the growth defect and increased cell viability in *CSA6^OE* mutant cells (Supplementary Fig. 7a, b). Next, we sought to characterize the

effect of SAC inactivation on the spindle integrity in *CSA6^OE* mutants. *CSA6^OE* resulted in the formation of an unconventional mitotic spindle (Fig. 3e) wherein it displayed a single focus of SPB (Tub4-GFP), colocalizing with a single focus of MTs (Tub1-mCherry). We speculated two possibilities that may lead to the single focus of Tub4: a) a defect in the process of SPB duplication or b) a delay in the separation of duplicated SPBs. Fluorescence microscopy analysis revealed that SAC inactivation in *CSA6^OE* mutant drastically increased the percentage of large-budded cells (from ~30% to ~68%) with two separated SPB foci (Tub4-GFP) (Supplementary Fig. 7c). These results ruled out the possibility of

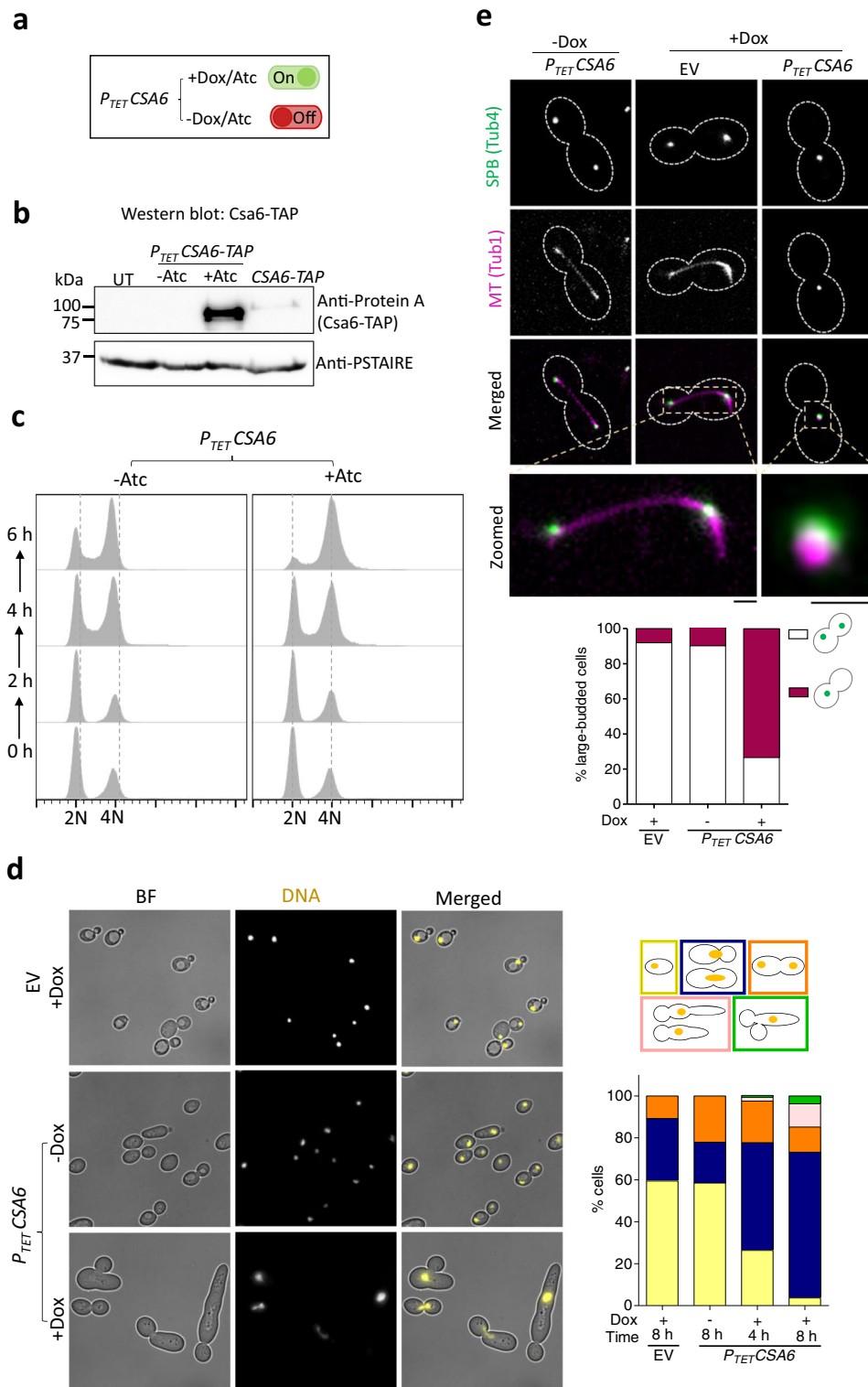

an unduplicated SPB in $CSA6^{OE}$ mutant cells and hinted at the importance of cellular Csa6 levels for proper SPB separation and chromosome segregation in *C. albicans*.

We next determined the effect of inactivating SPOC in the cells overexpressing Csa6. For this, we generated a $CSA6^{OE}$ strain (CaPJ200) using the *bub2* null mutant (CaPJ110) as the parent strain and monitored nuclear division following Hoechst staining. Strikingly, we did not observe a bypass of G2/M arrest in $CSA6^{OE}$

mutant upon SPOC inactivation, indicated by a persistent population of large-budded cells with unsegregated nuclei (Supplementary Fig. 7d). In addition, a marginal growth improvement, if any, was observed upon Bub2 deletion in the $CSA6^{OE}$ strain (Supplementary Fig. 7e, f). Altogether, our results demonstrate that overexpression of Csa6 leads to a Mad2-mediated metaphase arrest due to a malformed spindle in *C. albicans*.

**Fig. 3 Overexpression of Csa6 alters the morphology of the mitotic spindle and leads to G2/M arrest in *C. albicans*. a** Atc/Dox-dependent functioning of the $P_{TET}$ promoter system for conditional overexpression of *CSA6*. **b** Western blot analysis using anti-Protein A antibodies confirmed overexpression of *CSA6-TAP* from the $P_{TET}$ promoter (CaPJ181), after 8 h induction in presence of Atc (3 μg/ml), in comparison to the uninduced culture (-Atc) or *CSA6-TAP* expression from its native promoter (CaPJ180); $N = 2$. PSTAIRE was used as a loading control. UT, untagged control (SN148). **c** Flow cytometric analysis of cell cycle displaying the cellular DNA content of *CSA6^{OE}* strain (CaPJ176) in presence or absence of Atc (3 μg/ml) at the indicated time intervals; $N = 3$. The gating strategy used for plotting flow cytometric cell cycle data is illustrated in Supplementary Fig. 13. **d** *Left*, microscopic images of Hoechst-stained EV (CaPJ170) and *CSA6^{OE}* strain (CaPJ176) after 8 h of growth under indicated conditions of Dox (50 μg/ml). BF, bright-field. Scale bar, 10 μm. *Right*, quantitation of different cell types at the indicated time-points; $n \geq 100$ cells. **e** *Top*, representative micrographs of spindle morphology in the large-budded cells of EV (CaPJ172) and *CSA6^{OE}* strain (CaPJ178) after 8 h of growth under indicated conditions of Dox (50 μg/ml). SPBs and MTs are marked by Tub4-GFP and Tub1-mCherry, respectively. Scale bar, 1 μm. *Bottom*, the proportion of the large-budded cells with indicated SPB phenotypes; $n \geq 100$ cells. Source data are provided as a Source Data file.

**Csa6 regulates the mitotic exit network and is essential for viability in *C. albicans*.** To further gain insights into the biological function of Csa6, we sought to generate a promoter shutdown mutant of *csa6 (CSA6^{PSD})*. For this, we deleted one of its alleles and placed the remaining one under the control of the *MET3* promoter[62] which gets repressed in presence of methionine (Met/M) and cysteine (Cys/C) (Fig. 5a). Western blot analysis confirmed the depletion of TAP-tagged Csa6 in *CSA6^{PSD}* mutant within 6 h of growth under repressive conditions (Fig. 5b). The inability of the *CSA6^{PSD}* mutant to grow in non-permissive conditions confirmed the essentiality of Csa6 for viability in *C. albicans* (Fig. 5c, Supplementary Fig. 8a). Subsequently, we analyzed the cell cycle profile (Fig. 5d) and nuclear division dynamics (Fig. 5e) in the *CSA6^{PSD}* strain after a specific period of incubation in either permissive or non-permissive conditions. Strikingly, Csa6 depletion, as opposed to its overexpression, resulted in cell cycle arrest at the late anaphase/telophase stage, indicated by an increasing proportion of large-budded cells, possessing segregated nuclei and 4N DNA content (Fig. 5d, e). Additionally, we observed cells with more than two nuclei, elongated-budded cells and other complex phenotypes upon Csa6 depletion (Fig. 5e, Supplementary Fig. 8b). The *CSA6^{PSD}* mutant resumed growth without losing viability when shifted from non-permissive to permissive conditions (Supplementary Fig. 8c, d), indicating the arrest phenotype associated with the depletion of Csa6 is largely reversible. This is further supported by an increase in Csa6 levels upon switching growth conditions from non-permissive to permissive media (Supplementary Fig. 8e). While CENP-A/Cse4 remained localized to centromeres in *CSA6^{PSD}* mutant as revealed by the fluorescence microscopy (Supplementary Fig. 9a), an increase in the cellular levels of Cse4 was observed in *CSA6^{PSD}* mutant by western blot analysis (Supplementary Fig. 9b). An increase in Cse4 levels could be an outcome of Cse4 loading at anaphase in *C. albicans*[63,64]. Finally, we analyzed the integrity of the mitotic spindle, as mentioned previously, in *CSA6^{PSD}* mutant. We noticed the mean length of the anaphase mitotic spindle in Csa6-depleted cells was significantly higher (~11 μm) than that of the cells grown under permissive conditions (~6 μm), indicating a hyper-elongated aberrant mitotic spindle structure in the *CSA6^{PSD}* mutant (Fig. 5f).

A close link between anaphase arrest, hyper-elongated mitotic spindles and an inactive mitotic exit network (MEN) have been established before[41,65,66]. Localized at the SPB, the MEN is a signaling cascade in *S. cerevisiae* that triggers cells to come out of mitosis and proceed to cytokinesis (Fig. 6a)[67]. We speculated the anaphase arrest in *CSA6^{PSD}* mutant could be a result of an inactive MEN signaling. To determine this, we sought to bypass the anaphase arrest associated with Csa6 depletion by overexpressing *SOL1*, the CDK inhibitor and the Sic1 homolog in *C. albicans*[68] (Fig. 6b), using the inducible $P_{TET}$ system mentioned previously (Fig. 6c). The conditional overexpression of Protein A-tagged Sol1 upon addition of Atc was

verified by western blot analysis (Fig. 6d). *SOL1^{OE}* in association with Csa6 depletion allowed cells to exit mitosis but not cytokinesis, as evidenced by the formation of chains of cells with >4N DNA content (Fig. 6e, f). To further examine the role of Csa6 in mitotic exit, we analyzed the localization of a MEN component, Tem1, a GTPase that is known to initiate MEN signaling[40,69–71]. In *C. albicans*, Tem1 localizes to SPBs in a cell-cycle-regulated manner and is essential for cell viability[40]. Fluorescence microscopy revealed that while Tem1 is localized to both the SPBs in anaphase under permissive conditions (Fig. 6g) as observed earlier[40], a high percentage of Csa6-depleted cells (~78%) had Tem1 localized to only one of the two SPBs (Fig. 6g), suggesting an important role of Csa6 in regulating mitotic exit in *C. albicans*. In *S. cerevisiae*, a two-component GTPase-activating protein (GAP) complex consisting of Bub2 and Bfa1 is known to prevent mitotic exit by stimulating Tem1 GTPase activity[67]. To determine if Csa6 functions further upstream of Tem1, we sought to delete both copies of Bub2 in *CSA6^{PSD}* mutant as Bub2 but not Bfa1 carries a conserved GAP domain[72,73]. Strikingly, we observed a better growth of *CSA6^{PSD}* mutant under non-permissive conditions in the absence of Bub2 (Supplementary Fig. 10a). On the other hand, the growth defects of the Tem1 conditional mutant in *C. albicans*[40] were not rescued upon Bub2 deletion (Supplementary Fig. 10b). This is somewhat comparable to the deletion of Bub2 in *S. cerevisiae* that was shown to suppress the growth defect of a mitotic exit mutant *cdc5-1*[74] but not the *tem1-3* conditional mutant[73]. These results suggest that Csa6 functions at the proximal end of the MEN signaling pathway. Altogether, our results demonstrate that Csa6 is required for mitotic exit and thus essential for viability in *C. albicans*.

Considering the role of Csa6 is to promote mitotic exit by timely inactivation of Cdk1, overexpression of Csa6 is expected to inactivate Cdk1 at an earlier cell cycle stage. To test this hypothesis, we sought to monitor levels of the major mitotic cyclin Clb2[50,75] in the *CSA6^{OE}* mutant. For this, we first measured the Clb2 levels either at the early S (hydroxyurea-arrested) or at the G2/M (nocodazole-arrested) stage in cells carrying EV. These levels were compared with the Clb2 levels of the *CSA6^{OE}* strain arrested at the G2/M stage following Atc treatment (Fig. 3d, Supplementary Fig. 11a, b). While we confirmed lower levels of Clb2 in the early S-phase as reported previously[50], we found no significant difference in Clb2 levels between G2/M-arrested cells of EV and Atc-treated cells of the *CSA6^{OE}* strain (Supplementary Fig. 11a, b). These results suggest that Csa6 overexpression does not cause any major alterations in the mitotic cyclin Clb2 levels at the G2/M stage of the cell cycle.

**Csa6 of *Candida dubliniensis* and *Candida tropicalis* functionally complements Csa6 of *C. albicans*.** To further elucidate the intra-species function and localization of Csa6, we decided to ectopically express Csa6 of another CUG-Ser clade species,

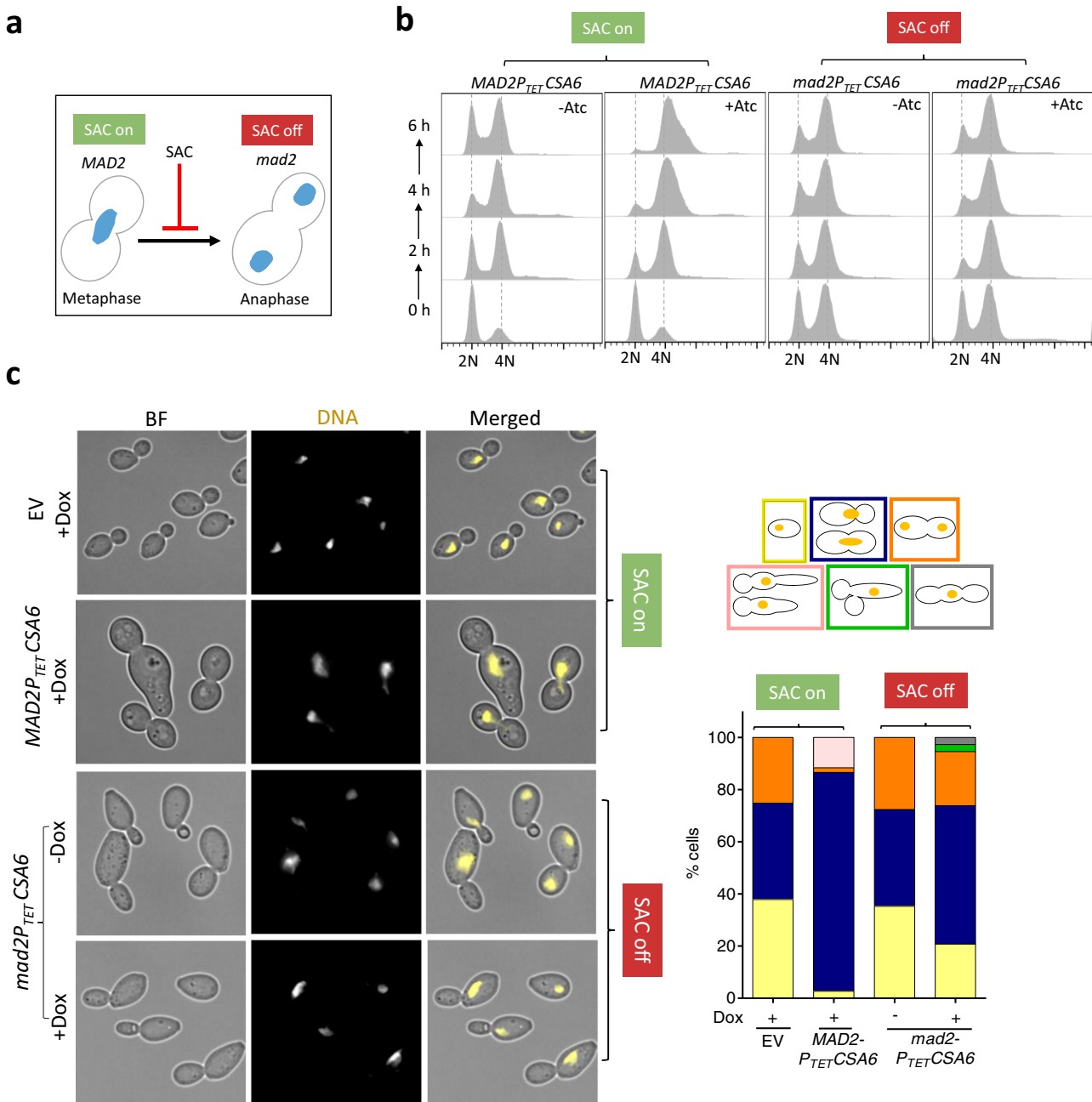

**Fig. 4 The G2/M cell cycle arrest in the *CSA6^OE* mutant is mediated by Mad2. a** The G2/M arrest posed by SAC in response to an improper chromosome-spindle attachment is relieved in the absence of Mad2, allowing cells to transit from metaphase to anaphase. **b** Flow cytometric DNA content analysis in CaPJ176 (*MAD2CSA6^OE*) and CaPJ197 (*mad2CSA6^OE*) at the indicated times, in presence or absence of Atc (3 μg/ml); N = 3. **c** *Left*, microscopic images of CaPJ170 (EV), CaPJ176 (*MAD2CSA6^OE*) and CaPJ197 (*mad2CSA6^OE*) following Hoechst staining, after 8 h of growth under indicated conditions of Dox (50 μg/ml). Scale bar, 10 μm. *Right*, quantitation of the indicated cell types; n ≥ 100 cells.

*Candida dubliniensis* (CdCsa6) in *C. albicans*. *C. dubliniensis* is a human pathogenic budding yeast that shares a high degree of DNA sequence homology with *C. albicans* and possesses unique and different centromere DNA sequences on each of its eight chromosomes[76,77]. Upon protein sequence alignment, we found that CdCsa6 (*ORF Cd36_16290*) is 79% identical to Csa6 of *C. albicans* (CaCsa6) (Fig. 7a). The ectopic expression of GFP-tagged CdCsa6 in *C. albicans* was carried out using the replicative plasmid pCdCsa6-GFP-ARS2 (Supplementary Fig. 12a), which contains the autonomously replicating sequence (ARS) of *C. albicans*[78]. Although unstable when present in an episomal form, ARS plasmids, upon spontaneous integration into the

genome, can propagate stably over generations[79]. Fluorescence microscopy of integrated pCdCsa6-GFP-ARS2 revealed that similar to CaCsa6, CdCsa6 localizes constitutively to the SPBs in *C. albicans* (Fig. 7b), further supporting Csa6's evolutionarily conserved role in regulating mitotic spindle and mitotic exit in *C. albicans*. We next asked if CdCsa6 can functionally complement CaCsa6. For this, we ectopically expressed CdCsa6 in the *CSA6^PSD* strain. Strikingly, the ectopic expression of CdCsa6 rescued the growth defect associated with *CSA6^PSD* mutant under non-permissive conditions, indicating CdCsa6 can functionally complement CaCsa6 (Fig. 7c).

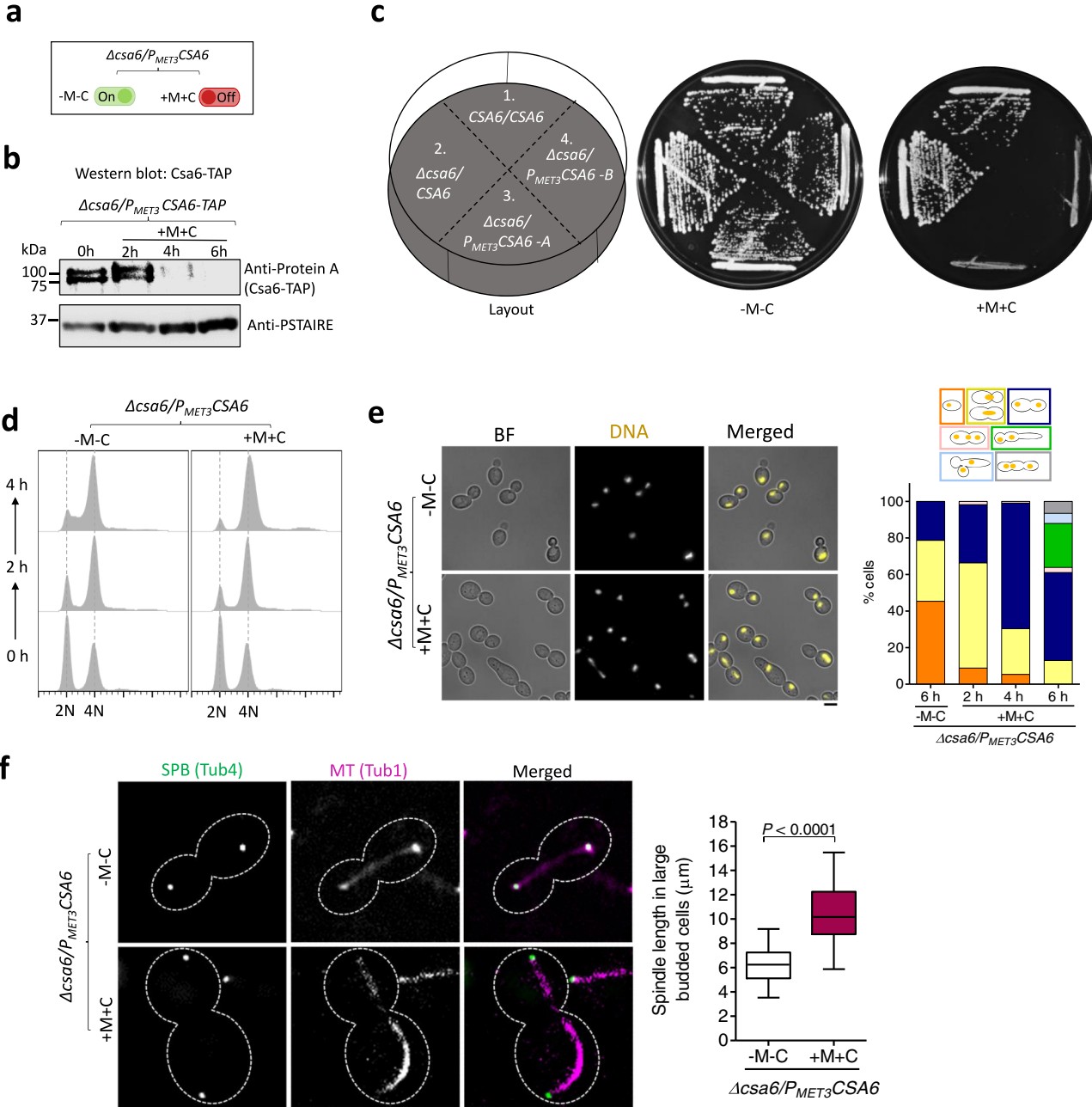

**Fig. 5 Csa6 depletion causes late anaphase/telophase arrest with a hyper-extended mitotic spindle in *C. albicans*. a** The *MET3* promoter system for depleting cellular levels of Csa6. The *MET3* promoter can be conditionally repressed in presence of methionine (Met/M) and cysteine (Cys/C). **b** Western blot analysis using anti-Protein A antibodies revealed time-dependent depletion of Csa6-TAP in *CSA6$^{PSD}$* strain (CaPJ212), grown under repressive conditions (YPDU + 5 mM Met and 5 mM Cys) for indicated time interval; $N = 2$. **c** Csa6 is essential for viability in *C. albicans*. Strains with indicated genotypes, (1) SN148, (2) CaPJ209, (3 and 4) CaPJ210 (two transformants) were streaked on agar plates with permissive (YPDU-Met-Cys) or repressive (YPDU + 5 mM Met and 5 mM Cys) media and incubated at 30 °C for two days. **d** Cell cycle analysis of CaPJ210 (*CSA6$^{PSD}$*) by flow cytometry under permissive (YPDU-Met-Cys) and repressive conditions (YPDU + 5 mM Met and 5 mM Cys) at the indicated time intervals; $N = 3$. **e** *Left*, microscopic images of Hoechst stained CaPJ210 (*CSA6$^{PSD}$*) cells grown under permissive (YPDU-Met-Cys) or repressive (YPDU + 5 mM Met and 5 mM Cys) conditions for 6 h. BF bright-field. Scale bar, 5 μm. *Right*, quantitation of different cell types at the indicated time-points; $n \geq 100$ cells. **f** *Left*, micrograph showing Tub4-GFP and Tub1-mCherry (representing mitotic spindle) in the large-budded cells of CaPJ211 (*CSA6$^{PSD}$*) after 6 h of growth under permissive (YPDU-Met-Cys) or repressive (YPDU + 5 mM Met and 5 mM Cys) conditions. Scale bar, 3 μm. *Right*, quantitation of the distance between the two SPBs, along the length of the MT (representing spindle length), in large-budded cells of CaPJ211 (*CSA6$^{PSD}$*) under permissive ($n = 32$) or repressive ($n = 52$) conditions. Box plots include the median line, the box denotes the interquartile range (IQR), whiskers extend down to the minimum value and up to the maximum value. Paired *t*-test, one-tailed, *P*-value shows a significant difference ($P < 0.0001$). Source data are provided as a Source Data file.

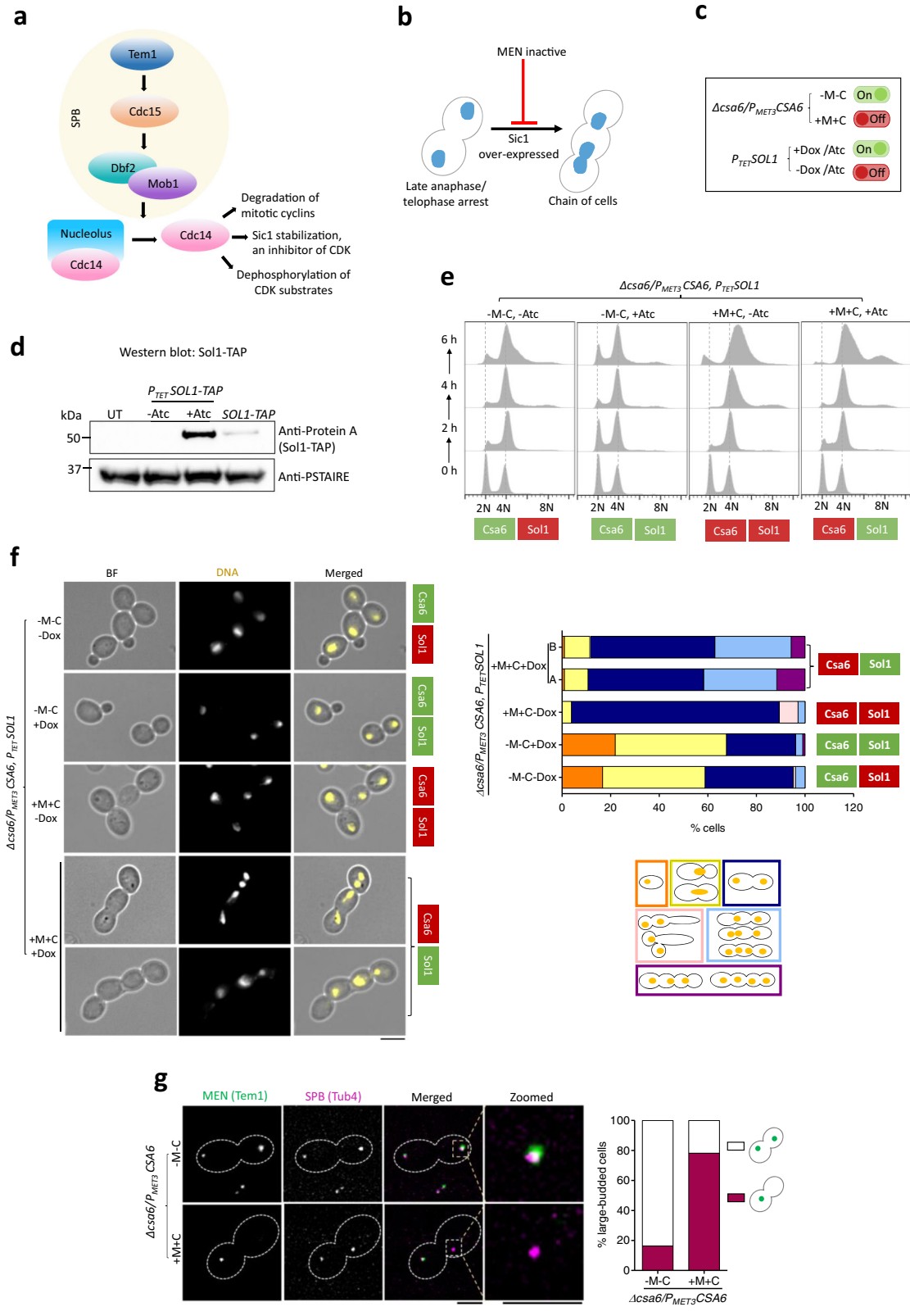

To further elucidate the function of Csa6 across CUG clades species, we ectopically expressed GFP-tagged Csa6 of two other *Candida* species, *Candida tropicalis* and *Candida parapsilosis*, in *C. albicans*. The putative Csa6 orthologs are poorly conserved in both *C. tropicalis* (CtCsa6) (Fig. 7d) and *C. parapsilosis* (CpCsa6) (Supplementary Fig. 12b). Incidentally, CtCsa6, but not CpCsa6, functionally complemented CaCsa6 functions in *C. albicans*

(Fig. 7e). Similar to CaCsa6 and CdCsa6, CtCsa6 is also localized to the SPBs throughout the cell cycle in *C. albicans* (Fig. 7f). In contrast, only a small percentage of cells had CpCsa6 localized to the SPBs in *C. albicans* (Supplementary Fig. 12c). In the majority of cells, CpCsa6 was neither localized to the SPBs nor showed any detectable fluorescence signals (Supplementary Fig. 12c). This suggests Csa6 is functionally conserved in species closely related

**Fig. 6 Csa6 is required for mitotic exit in *C. albicans*. a** The MEN components in *S. cerevisiae*. At SPB, Nud1 acts as a scaffold. The ultimate target of the MEN is to activate Cdc14 phosphatase, which remains entrapped in the nucleolus in an inactive state until anaphase. Cdc14 release brings about mitotic exit and cytokinesis by promoting degradation of mitotic cyclins, inactivation of mitotic CDKs through Sic1 accumulation and dephosphorylation of the CDK substrates[67]. **b** Inhibition of the MEN signaling prevents cells from exiting mitosis and arrests them at late anaphase/telophase. Bypass of cell cycle arrest due to the inactive MEN, viz. by overexpression of Sic1-a CDK inhibitor, results in the chain of cells with multiple nuclei[111,112]. **c** A combination of two regulatable promoters, $P_{TET}$ and $P_{MET3}$, was used to overexpress *C. albicans* homolog of Sic1, called *SOL1* (Sic one-like), in Csa6-depleted cells. The resulting strain, CaPJ215, can be conditionally induced for both *SOL1* overexpression upon Atc/Dox addition and Csa6 depletion upon Met (M)/Cys (C) addition. **d** Protein A western blot analysis showed increased levels of Sol1 (TAP-tagged) in the *SOL1^OE^* mutant (CaP217, $P_{TET}SOL1$-TAP) after 6 h induction in presence of Atc (3 µg/ml) in comparison to the uninduced culture (-Atc) or *SOL1* expression from its native promoter (CaPJ216, *SOL1*-TAP); $N = 2$. PSTAIRE was used as a loading control. UT untagged control (SN148). **e** Flow cytometric analysis of cell cycle progression in CaPJ215 at indicated time intervals under various growth conditions, as indicated; $N = 3$. Dox: 50 µg/ml, Met: 5 mM, Cys: 5 mM. **f** *Left*, Hoechst staining of CaPJ215 after 6 h of growth under indicated conditions of Dox (50 µg/ml), Met (5 mM) and Cys (5 mM); $n \geq 100$ cells. BF bright-field. Scale bar, 5 µm. *Right*, percent distribution of the indicated cell phenotypes; $n \geq 100$ cells. **g** *Left*, co-localization analysis of Tem1-GFP and Tub4-mCherry in large-budded cells of CaPJ218 (*CSA6^PSD^*) under permissive (YPDU-Met-Cys) or repressive conditions (YPDU + 5 mM Met and 5 mM Cys). Scale bar, 3 µm. *Right*, the proportion of the large-budded cells with indicated Tem1 phenotypes; $n \geq 100$ cells. Source data are provided as a Source Data file.

to *C. albicans* such as *C. dubliniensis* and *C. tropicalis* but its function might have diverged in more distant species such as *C. parapsilosis*.

## Discussion

In this study, we carried out an extensive screen to identify genes that contribute to genome stability in *C. albicans* by generating and analyzing a library of more than a thousand overexpression strains. Our screen identified six regulators of chromosome stability including Csa6, a protein of unknown function. Besides revealing the sub-cellular localization of Csa6 at the SPBs, we demonstrated the apparent complexity of the cellular role of Csa6. Whereas Csa6 overexpression arrests the cells at the SAC with an improperly assembled spindle, its depletion leads to a terminal defect later in the cell cycle and arrests the cells in late anaphase/telophase. Finally, subcellular localization and complementation analysis revealed functional conservation of Csa6 across some of the pathogenic *Candida* species.

The six *CSA* genes identified in the study are important for mitotic progression in *C. albicans* or related yeast species (Fig. 8a, b). We, therefore, believe that overexpression of these six genes will induce CIN in *C. albicans*, regardless of the choice of the chromosome under investigation. Indeed, the identification of two *CSA* genes, *CSA2^ASE1^* and *CSA5^BFA1^*, that were earlier reported as CIN genes[13,14], further validates the power of the screening approach and the methods presented in this study. The respective overexpression phenotypes of these two genes in *C. albicans* were found to be like those in *S. cerevisiae*, suggesting that their functions might be conserved in these distantly related yeast species. In *S. cerevisiae*, Ase1 acts as an MT-bundling protein, required for spindle elongation and stabilization during anaphase[80,81] (Fig. 8a). Hence, increased CIN upon *ASE1* overexpression might be an outcome of premature spindle elongation and improper KT-microtubule attachments[81,82]. Bfa1, on the other hand, is a key component of the Bub2-Bfa1 complex, involved in SPOC activation[60], and a negative regulator of mitotic exit[83] (Fig. 8a). In *S. cerevisiae*, *BFA1* overexpression prevents Tem1 from interacting with its downstream effector protein Cdc15, thus inhibiting MEN signaling and arresting cells at the anaphase[74]. In our screen, a B-type mitotic cyclin Clb4 (*CSA1*), and a kinesin-related motor protein Kip2 (*CSA3*) (Fig. 8a), were found to increase CIN upon overexpression, primarily via non-chromosomal loss events. *C. albicans* Clb4 acts as a negative regulator of polarized growth[50] and is the functional homolog of *S. cerevisiae* Clb5[84], required for the entry into the S-phase[85]. Increased CIN upon *CSA1^CLB4^* overexpression, is thus consistent with its role in S-phase initiation. The function of Kip2, however, is yet to be characterized in *C. albicans*. In *S. cerevisiae*, Kip2

functions as an MT polymerase[86], with its overexpression leading to hyperextended MTs and defects in SPB separation[87]. The associated CIN observed upon *CSA3^KIP2^* overexpression in *C. albicans* is in line with its function in nuclear segregation.

Mcm7, another *CSA* gene (*CSA4*) identified in this study, is a component of the highly conserved Mcm2-7 helicase complex, essential for eukaryotic DNA replication initiation and elongation[88] (Fig. 8a). While Mcm7 depletion arrests cells at S phase[89], the effect of *MCM7* overexpression on genomic integrity is comparatively less explored. Especially, several cancerous cells have been shown to overexpress Mcm7[90–92], with its elevated levels increasing the chances of relapse and local invasions[90]. In this study, we found that overexpression of *MCM7*, in contrast to Mcm7 depletion, arrested cells at the G2/M stage. One possibility is that increased Mcm7 levels interfered with DNA replication during the S phase, resulting in DNA damage or accumulation of single-stranded DNA, thus activating the *RAD9*-dependent cell cycle arrest at the G2/M stage[93,94]. In a recent study from our laboratory, Mcm7 has been identified as a subunit of the kinetochore interactome in a basidiomycete yeast *Cryptococcus neoformans*[95]. Another subunit of the Mcm2-7 complex, Mcm2, is involved in regulating the stability of centromeric chromatin in *C. albicans*[64]. Considering the growing evidence of the role of Mcm2-7 subunits beyond their canonical, well-established roles in DNA replication, the serendipitous identification of Mcm7 as a regulator of genome stability in our screen is striking.

We performed an in-depth analysis of Csa6, a critical regulator of cell cycle progression identified from our screen (Fig. 8b, c). Our results revealed that overexpression of *CSA6* leads to an unconventional mitotic spindle formation and SAC-dependent G2/M cell cycle arrest (Fig. 8c) in *C. albicans*. Likewise, in budding yeast, depletion of Scm3[96] and several *mcm* mutants[97,98] under restrictive conditions are shown to cause both cell cycle arrest and chromosome segregation defects. In humans, chromosome mis-segregation or aneuploidy can lead to further genomic instability that ultimately causes cell cycle arrest[99]. Whether cell cycle arrest upon Csa6 overexpression is a cause or a consequence of chromosomal instability remains an enigma. While *mad2* deletion indicated that SPB duplication and separation of duplicated SPBs is unperturbed in *CSA6* over-expressing cells, what exactly triggered the activation of SAC in these cells remains to be determined. Recent studies on human cell lines have shown that failure in the timely separation of the centrosomes promotes defective chromosome-MT attachments and may lead to chromosome lagging if left uncorrected by the cellular surveillance machinery[100–102]. Along the same lines, we posit that a delay in SPB separation, mediated by overexpression of Csa6, leads to increased instances of improper chromosome-

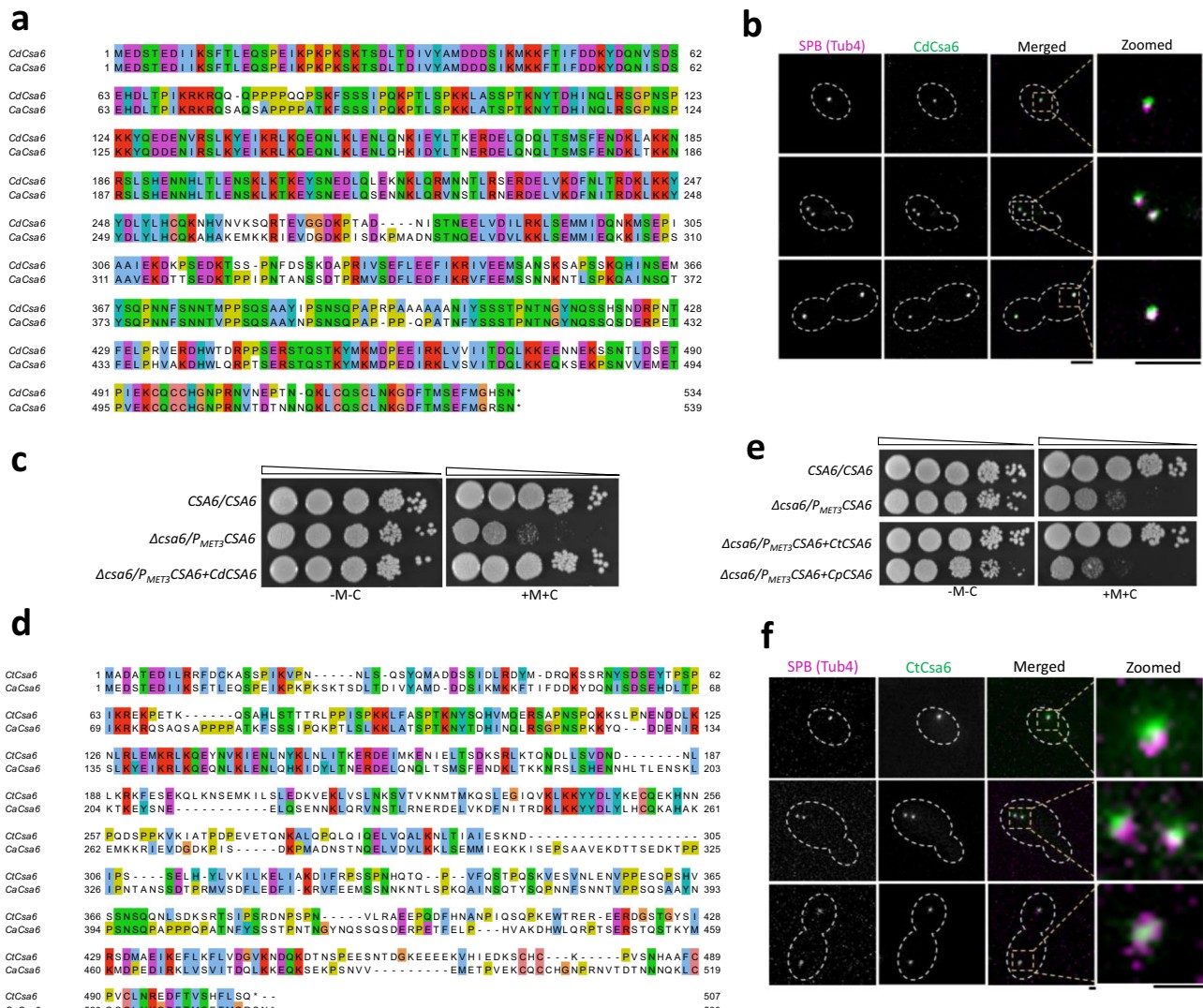

**Fig. 7 Functional complementation of CaCsa6 by its homologs in related species. a** Pair-wise alignment of amino acid sequences of Csa6 proteins in *C. albicans* (CaCsa6) and *C. dubliniensis* (CdCsa6) by Clustal Omega, visualized using Jalview. **b** CdCsa6 localizes to the SPB. Representative micrographs showing CdCsa6GFP localization at different cell cycle stages in CaPJ300. Tub4mCherry was used as an SPB marker. Scale bar, 3 μm. **c** CdCsa6 functionally complements CaCsa6. Ten-fold serial dilutions of SN148 (*CSA6/CSA6*), CaPJ301(*CSA6^PSD*) and CaPJ302 (*CSA6^PSD* expressing *CdCSA6*), starting from 10^5 cells were spotted on agar plates with permissive (YPDU-Met-Cys) or repressive (YPDU + 5 mM Met and 5 mM Cys) media and incubated at 30 °C for two days; *N* = 3. **d** Pair-wise protein sequence alignment of Csa6 of *C. tropicalis* (CtCsa6) and CaCsa6. **e** CtCsa6 but not CpCsa6 (Csa6 of *C. parapsilosis*) functionally complements CaCsa6. Spot dilution analysis of SN148 (*CSA6/CSA6*), CaPJ301(*CSA6^PSD*), CaPJ303 (*CSA6^PSD* expressing *CtCSA6*) and CaPJ304 (*CSA6^PSD* expressing *CpCSA6*). Ten-fold serial dilutions, starting from 10^5 cells were spotted on permissive or repressive media; *N* = 3. **f** Representative micrographs showing constitutive localization of CtCsa6 at the SPBs in CaPJ303 (*CSA6^PSD* expressing *CtCSA6*) under permissive conditions. SPBs are marked using Tub4mCherry. Scale bar, 1 μm.

MT attachments, leading to SAC activation and an indefinite arrest at the metaphase stage. Future studies on the SPB structure-function and composition in *C. albicans* should reveal how Csa6 regulates SPB dynamics in this organism.

In contrast to its overexpression, Csa6 depleted cells failed to exit mitosis and remained arrested at the late anaphase/telophase stage (Fig. 8c). We further linked the mitotic exit failure in Csa6 depleted cells with the defective localization of Tem1, an upstream MEN protein. While the hierarchy of MEN components, starting from the MEN scaffold Nud1, an SPB protein, to its ultimate effector Cdc14 is well established in *S. cerevisiae*[67], the existence of a similar hierarchy in *C. albicans* needs to be investigated. In addition, several lines of evidence suggest that MEN in *C. albicans* may function differently from *S. cerevisiae*: (a) Unlike *S. cerevisiae*, *C. albicans* Cdc14 is non-essential for

viability with its deletion affecting cell separation[103]. (b) Cdc14 is present in the nucleoplasm for most of the cell cycle in contrast to its nucleolar localization in *S. cerevisiae*[103]. (c) *C. albicans* Dbf2 is required for proper nuclear segregation, actomyosin ring contraction, and cytokinesis[39]. A recent study involving the identification of the Cdc14 interactome in *C. albicans*[104] found only a subset of proteins (0.2%) as physical or genetic interactors in *S. cerevisiae*, suggesting the divergence of Cdc14 functions in *C. albicans*. Hence, further investigations of MEN functioning in *C. albicans* are required to understand its divergence from *S. cerevisiae* and the mechanism by which Csa6 regulates mitotic exit in *C. albicans* and related species. We explored the possibility that Csa6 might function to promote mitotic exit via Cdk1 inactivation. In such a scenario, Csa6 overexpression early in the cell cycle may halt mitotic progression by preventing the proper raise

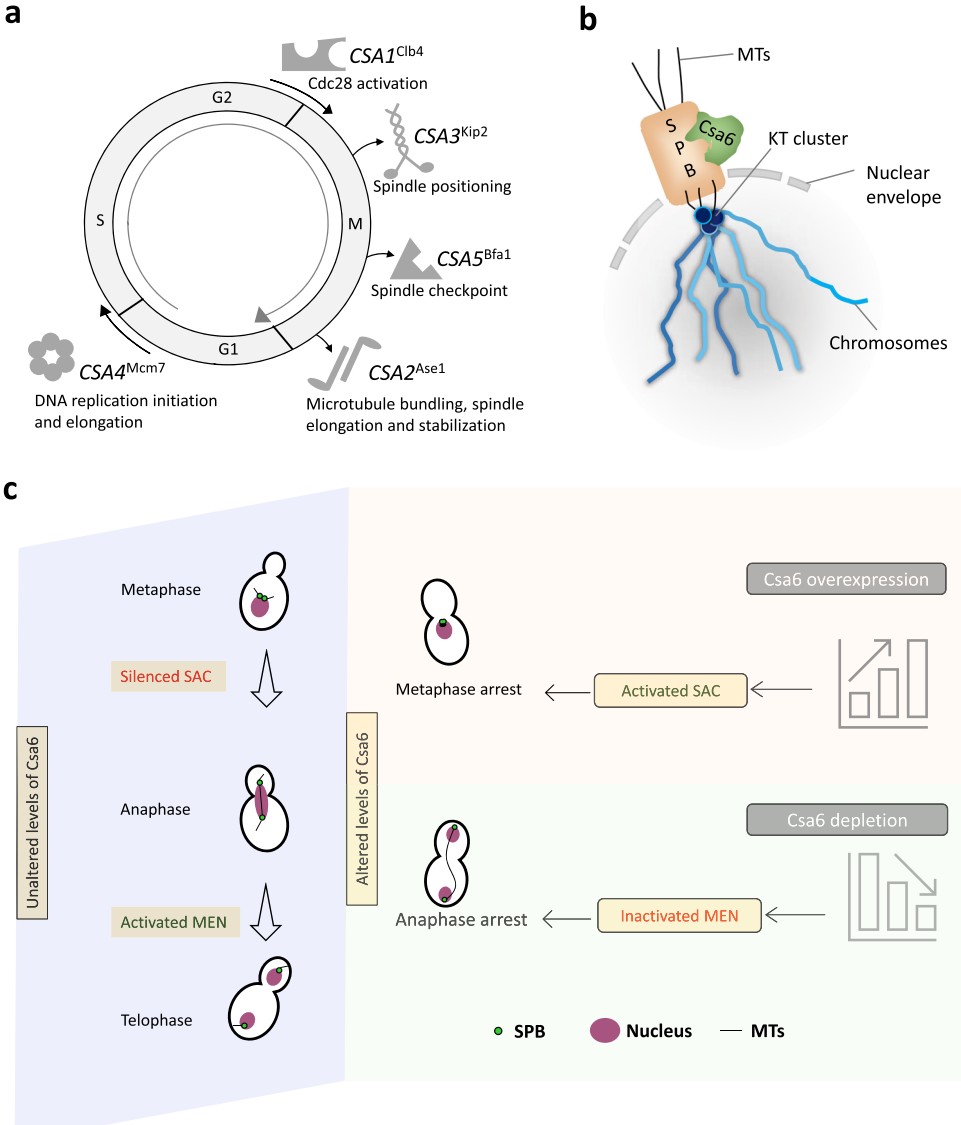

**Fig. 8 Csa6 levels are fine-tuned at various stages of the cell cycle to ensure both mitotic progression and mitotic exit in *C. albicans*. a** A diagram illustrating the functions of the identified *CSA* genes except for *CSA6* in various phases and phase transitions of the cell cycle. **b** Schematic depicting the approximate position of Csa6 with respect to SPB and KT. In *C. albicans*, SPBs and clustered KTs remain in close proximity throughout the cell cycle, while Csa6 remains constitutively localized to the SPBs. **c** A model summarizing the effects of overexpression or depletion of Csa6 in *C. albicans*. A wild-type cell with unperturbed Csa6 levels progresses through the mitotic cell cycle. Overexpression of *CSA6* alters the mitotic spindle dynamics, which might lead to improper KT-MT attachments, prompting SAC activation and G2/M arrest. In contrast, decreased levels of Csa6 inhibit the MEN signaling pathway, probably by affecting Tem1 recruitment to the SPBs, resulting in cell cycle arrest at the late anaphase/telophase stage.

of Cdk1 activity. While we found unaltered levels of Clb2 upon Csa6 overexpression (Fig. S9), there is a possibility of alternate mechanisms, other than cyclin degradation, leading to Cdk1 inactivation in the $CSA6^{OE}$ mutant. Taken together, our results indicate the importance of Csa6 in the mitotic cell cycle progression at least in two critical stages, the metaphase-anaphase transition and at the mitotic exit. In addition, the constitutive localization of Csa6 to the SPBs strengthens the link between SPB-related functions and Csa6 in *C. albicans* (Fig. 8b, c).

The phylogenetic analysis of Csa6 revealed that it is only present in a group of fungal species, belonging to the CUG-Ser clade. Combined with its essential cell-cycle-related functions, it is intriguing to determine whether emergence of Csa6 is required to keep the pace of functional divergence in the regulatory mechanisms of cell cycle progression in these *Candida* species. While we demonstrated that Csa6 of *C. dubliniensis* and *C.*

*tropicalis* functionally complements Csa6 of *C. albicans*, whether Csa6 can complement the function of any known SPB-associated protein remains to be investigated. A recent study shows that around 50 essential genes, including Csa6, are only present in a group of *Candida* species (see Dataset 5 in[105]). Identification and functional characterization of these genes in the future will aid in developing clade-specific antifungal therapies[105]. In this study, we have analyzed only a part of the *C. albicans* ORFeome for their roles in genome maintenance. Further screening of the remaining overexpression ORFs will provide a complete network of the molecular pathways regulating genome stability in human fungal pathogens.

## Methods

**Strains, plasmids and primers**. Information related to strains, plasmids and primers used in this study is available in the Supplementary Information. Strategies

used for generating strains and plasmids are mentioned in Supplementary Methods. The list of all the yeast strains, primers and plasmids used in this study are mentioned in Supplementary Tables 3, 4 and 5, respectively.

**Media and growth conditions.** *C. albicans* strains were routinely grown at 30 °C in YPD (1% yeast extract, 2% peptone, 2% dextrose) medium supplemented with uridine (0.1 μg/ml) or complete medium (CM, 2% dextrose, 1% yeast nitrogen base and auxotrophic supplements) with or without uridine (0.1 μg/ml) and amino acids such as histidine, arginine, leucine (0.1 μg/ml). Solid media were prepared by adding 2% agar. For the selection of transformants, nourseothricin and hygromycin B (hyg B) were used at a final concentration of 100 μg/ml and 800 μg/ml, respectively, in the YPDU medium.

Overexpression of genes from the tetracyline inducible promoter ($P_{TET}$) was achieved by the addition of anhydrotetracycline (Atc, 3 μg/ml) or doxycycline (Dox, 50 μg/ml) in YPDU medium at 30 °C[48] in the dark as Atc and Dox are light-sensitive. The $CSA6^{PSD}$ strains were grown at 30 °C either in permissive (YPDU) or nonpermissive (YPDU + 5 mM methionine (M) + 5 mM cysteine (C)) conditions of the *MET3* promoter[62,64]. *E. coli* strains were cultured at 30 °C or 37 °C in Luria-Bertani (LB) medium or 2YT supplemented with ampicillin (50 μg/ml or 100 μg/ml), chloramphenicol (34 μg/ml), kanamycin (50 μg/ml) and tetracycline (10 μg/ml). Solid media were prepared by adding 2% agar. Chemically competent *E. coli* cells were prepared according to Chung et al.[106].

**Flow cytometry analysis.** Cultures of overexpression strains following 8 h of induction in YPDU + Atc and overnight recovery in the YPDU medium alone, were diluted in 1x phosphate-buffered saline (PBS) and analyzed (~$10^6$ cells) for the BFP/GFP marker by flow cytometry (FACSAria III, BD Biosciences) at a rate of 7000-10,000 events/s. We used 405- and 488-nm lasers to excite the BFP and GFP fluorophores and 450/40 and 530/30 filters to detect the BFP and GFP emission signals, respectively.

**Primary and secondary overexpression screening.** To detect CIN at the BFP/GFP locus upon $P_{TET}$ activation, overnight grown cultures of *C. albicans* over-expression strains were reinoculated in CM-His-Arg to ensure all cells contained *BFP-HIS1* or *GFP-ARG4*. To measure the loss of BFP/GFP signals in 96-well plates, a $CDC20^{OE}$ mutant was used as a positive control. The primary selection of the overexpression mutants with increased $BFP^+GFP^-$ and $BFP^-GFP^+$ cells was done by determining the BFP/GFP loss frequency in EV. For this, we analyzed the flow cytometry density plots for 22 independent cultures of EV using the FlowJo software (FlowJo X 10.0.7r2). We observed a similar profile for all the cultures. We then defined gates for the $BFP^+GFP^-$ and $BFP^-GFP^+$ fractions of cell population in one of the EV samples and applied these gates to the rest of EV samples. The mean frequency of $BFP^+GFP^-$ and $BFP^-GFP^+$ cells in EV was calculated (Supplementary Table 1). Similar gates were applied to all 1067 overexpression strains analyzed for BFP/GFP markers and the frequency of $BFP^+GFP^-$ and $BFP^-GFP^+$ cells for each strain was determined (Supplementary Data 1). The overexpression mutants, in which the BFP/GFP loss frequency was ≥2-fold than EV, were selected for further analysis (Suplementary Table 2).

For secondary screening, the overexpression plasmids present in each of the overexpression strains, identified from the primary screen (23 out of 1067), were used to retransform the CSA reporter strain (CEC5201). The overexpression strains (23) were analyzed by flow cytometry to revalidate the loss of BFP/GFP signals. Overexpression strains displaying ≥2-fold higher frequency of $BFP^+GFP^-$ /$BFP^-GFP^+$ population than EV (6 out of 23) were monitored for any morphological transition by microscopy. As filamentous morphotype could distort the BFP/GFP loss analysis[47], we characterized the overexpression mutants exhibiting increased CIN at the BFP/GFP locus and filamentous growth (3 out of 6) by monitoring cell cycle progression. For this, we transformed the overexpression plasmids in CaPJ159 and analyzed the overexpression strains ($CSA4^{MCM7}$, $CSA5^{BFA1}$ and *CSA6*) for DNA content, nuclear segregation and SPB separation. The 6 genes identified from the secondary screen were verified for the correct *C. albicans* ORF by Sanger sequencing using a common primer PJ90. During the secondary screening, we also cultured overexpression mutants in YPDU without Atc and observed no differences between EV and uninduced (-Atc) cultures in terms of morphology and the BFP/GFP loss frequency.

**Cell sorting and marker analysis following a CIN event.** Overnight grown cultures of EV and overexpression mutants (*CDC20*, $CSA1^{CLB4}$, $CSA2^{ASE1}$ and $CSA3^{KIP2}$) were reinoculated in YPDU + Atc for 8 h and allowed to recover overnight in YPDU-Atc. The cultures were analyzed for BFP/GFP loss by flow cytometry followed by fluorescence-activated cell sorting (FACS) using a cell sorter (FACSAria III, BD Biosciences) at a rate of 10,000 events/s. Approximately 1500 cells from the $BFP^-GFP^+$ population were collected into 1.5-ml tubes containing 400 μl YPDU and immediately plated onto YPDU agar plates. Upon incubation at 30 °C for 2 days, both small and large colonies appeared, as reported earlier[47].

For marker analysis, we replica plated large or small colonies along with the appropriate control strains on CM-Arg, CM-His and YPDU + hyg B (800 μg/ml) and incubated the plates at 30 °C for 2 days. The colonies from CM-Arg plates were analyzed for BFP, GFP and RFP markers by flow cytometry. For this, overnight

grown cultures in YPDU were diluted in 1x PBS and 5000-10,000 cells were analyzed (FACSAria III, BD Biosciences). We used 405-, 488- and 561 nm lasers to excite the BFP, GFP and RFP fluorophores and 450/40, 530/30, 582/15 filters to detect the BFP, GFP and RFP emission signals, respectively.

**Cell cycle analysis.** Overnight grown cultures of *C. albicans* were reinoculated at an $OD_{600}$ of 0.2 in different media (as described previously) and harvested at various time intervals post-inoculation (as mentioned previously). The overnight grown culture itself was taken as a control sample (0 h) for all the experiments. Harvested samples were processed for propidium iodide (PI) staining as described before[33]. Stained cells were diluted to the desired cell density in 1× PBS and analyzed (≥30,000 cells) by flow cytometry (FACSAria III, BD Biosciences) at a rate of 250-1000 events/s. The output was analyzed using the FlowJo software (FlowJo X 10.0.7r2). We used 561-nm laser to excite PI and 610/20 filter to detect its emission signals.

**Fluorescence microscopy.** For nuclear division analysis in untagged strains, the *C. albicans* cells were grown overnight. The next day, the cells were transferred into different media (as mentioned previously) with a starting $O.D._{600}$ of 0.2, collected at various time intervals (as described previously) and fixed with formaldehyde (3.7%). Cells were pelleted and washed thrice with 1x PBS, and Hoechst dye (50 ng/ml) was added to the cell suspension before imaging. Nuclear division in Cse4-and Tub4-tagged strains was analyzed as described above, except the cells were not fixed with formaldehyde. For Tem1 and mitotic spindle localization, overnight grown cultures were transferred to different media (as mentioned previously) with a starting $O.D._{600}$ of 0.2 and were grown for 6 h or 8 h. Cells were then washed, resuspended in 1× PBS and imaged on a glass slide. Localization studies of each, Tub4, Spc110, Cmd1, CaCsa6, CdCsa6, CtCsa6 and CpCsa6 were carried out by washing the log phase grown cultures with 1x PBS (three times) followed by image acquisition.

The microscopy images were acquired using fluorescence microscope (Zeiss Axio Observer 7 equipped with Colibri 7 as the LED light source), 100× Plan Apochromat 1.4 NA objective, pco. edge 4.2 sCMOS. We used Zen 2.3 (blue edition) for image acquisition and controlling all hardware components. Filter set 92 HE with excitation 455–483 and 583–600 nm for GFP and mCherry, respectively, and corresponding emission was captured at 501–547 and 617–758 nm. Z sections were obtained at an interval of 300 nm. All the images displayed after the maximum intensity projection using ImageJ. Image processing was done using ImageJ. We used the cell counter plugin of ImageJ to count various cell morphologies in different mutant strains. Images acquired in the mCherry channel were processed using the subtract background plugin of ImageJ for better visualization.

**Protein preparation and western blotting.** Approximately 3 O.D.$_{600}$ equivalent cells were taken, washed with water once and resuspended in 12.5% TCA (tri-chloroacetic acid) and incubated at −20 °C overnight for precipitation. The cells were pelleted down and washed twice with ice-cold 80% acetone. The pellet was then allowed to air dry and finally resuspended in lysis buffer (0.1 N NaOH and 1% SDS and 5xprotein loading dye). Samples were boiled at 95 °C for 5-10 min and electrophoresed on a 10% SDS polyacrylamide gel. Gels were transferred to a nitrocellulose membrane by semi-dry method for 30 min at 25 V and blocked for an hour in 5% non-fat milk in 1x PBS. Membranes were incubated with a 1:5000 dilution of rabbit anti-Protein A or mouse anti-PSTAIRE in 2.5% non-fat milk in 1x PBS. Membranes were washed three times in 1x PBS-Tween (0.05%) and then exposed to a 1:10,000 dilution of either anti-mouse- or anti-rabbit-IgG horseradish peroxidase antibody in 2.5% non-fat milk in 1x PBS. Membranes were washed three times in 1x PBS-Tween (0.05%) and developed using the chemiluminescence method. Uncropped and unprocessed blots are shown in the Source Data file.

**Viability assays.** Overnight grown cultures of *C. albicans* were reinoculated in different media (as mentioned previously) with a starting $O.D._{600}$ of 0.2 and incubated for various time points (as described previously). Cells equivalent to O.D.$_{600}$ of 1 were then pelleted, washed, and resuspended in 1 ml water after which they were serially diluted and spotted on YPDU agar plates.

**Whole-genome sequence analysis.** To eliminate large debris and filamentous cells that could obstruct the tubing system of the cytometer, induced and non-induced cultures were filtered using BD Falcon™ Cell strainers. The MoFlo® Astrios™ flow cytometer was used to analyze and sort the cells of interest. For each culture, 500 mono-GFP cells were recovered in 400 μL of liquid YPD medium, plated immediately after cell sorting on four YPD agar plates and incubated at 30 °C for 48 h.

For DNA extraction and whole-genome sequencing, colonies were cultured in 3 ml of liquid YPD overnight at 30 °C, and DNA was extracted by following the manufacturer's protocol using the Qiagen QIAamp DNA minikit. The DNA was eluted in a total volume of 100 μl. The genomes were sent for whole-genome sequencing at Novogene, Illumina sequencing technology. Libraries were constructed using the yeast resequencing kit: Novogene NGS DNA Library Prep Set. The NovoSeq6000 platforms were used to generate 150 bp paired-ends reads.

For genome sequence analysis, sequences and genomic variations were analyzed as described previously[107,108]. Heterozygous SNVs were defined as positions where 15% or more of the calls showed one allele and 85% or less of the calls showed a second allele. Heterozygous SNP density maps were constructed by determining the number of heterozygous positions per 10-kb region and plotting each value.

**Statistical analysis and reproducibility**. Statistical significance of differences was calculated as mentioned in the figure legends with unpaired one-tailed *t*-test, paired one-tailed *t*-test, unpaired two-tailed *t*-test, paired two-tailed *t*-test or one-way ANOVA with Bonferroni posttest. *P*-values ≥ 0.05 were considered as non-significant (n.s.). *P*-values of the corresponding figures are mentioned, if significant. All analyses were conducted using GraphPad Prism version Windows v5.00. Micrographs shown in figures are representative of at least two independent experiments with similar results.

**Availability of materials**. Strains and plasmids are available from the corresponding authors upon reasonable request.

**Reporting Summary**. Further information on research design is available in the Nature Research Reporting Summary linked to this article.

## Data availability

Source data are provided with this paper: source data underlying Fig. 3b; Fig. 5b, f; Fig. 6d; Supplementary Fig. 1e; Supplementary Fig. 1g; Supplementary Fig. 3c–e; Supplementary Fig. 6d; Supplementary Fig. 8e; Supplementary Fig. 9b; and Supplementary Fig. 11a-b are provided in the Source Data file. Genome sequences have been deposited in the NCBI Sequence Read Archive under BioProject ID PRJNA842202. Publicly available databases used in the study include *Candida* Genome database (www.candidagenome.org), *Saccharomyces* Genome database (www.yeastgenome.org) and NCBI nonredundant protein database (https://blast.ncbi.nlm.nih.gov/Blast.cgi). Other data supporting the findings of this study are available from the corresponding authors upon request. Source data are provided with this paper.

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

## Acknowledgements
We thank members of the Sanyal and d'Enfert laboratories for their valuable suggestions and constructive criticisms. We thank the Munro group at University of Aberdeen and Mazel group at Institut Pasteur for their contribution to the establishment of over-expression plasmids that were used in this study, a work that will be reported elsewhere. We thank Dr. Arshad Desai for critical reading of the manuscript. We thank N. Varshney for constructing the plasmid pCse4-TAP-Leu. We thank L. Sreekumar for constructing pTub4-GFP-His cassette. Special thanks to K. Guin for sharing the raw files to generate the phylogenetic tree. We thank V. Sood and A. Das for generating the plasmid pCdCsa6-GFP-ARS2. We thank A.S. Amrutha for generating the strains CaPJ300 and CaPJ301. We thank H. Arya for generating the plasmid pCIp10-GFP-CtCsa6 and the strain CaPJ303. We acknowledge BioRender.com for the generation of Supplementary Fig. 4. We acknowledge N. Nala at the flow cytometry facility, JNCASR, for assisting flow cytometry and cell sorting experiments. We thank Pierre-Henri Commere from the Utechs CB, Institut Pasteur for flow cytometry sorting. The establishment of over-expression plasmids was supported by the Wellcome Trust [088858/Z/09/Z to CD]. This work was supported by a grant from the Indo French Centre for the Promotion of Advanced Research (CEFIPRA, Project no. 5703−2) as well as intramural funding from Institut Pasteur and Institut national de la recherche pour l'agriculture, l'alimentation et l'environnement (INRAE). This work is also supported by a Department of Bio-technology grant in Life Science Research, Education and Training at Jawaharlal Nehru Centre for Advanced Scientific Research (BT/INF/22/SP27679/2018). CEFIPRA also aided in the travel of PJ, KS and CD between the Sanyal and d'Enfert laboratories. PJ acknowledges intramural funding from JNCASR. AD and TP were supported by the CEFIPRA grant. K.S. acknowledges the financial support of JC Bose National Fellowship (Science and Engineering Research Board, Govt. of India, JCB/2020/000021) and intra-mural funding from JNCASR.

## Author contributions
Conceptualization: K.S., C.D., P.J., M.L. Methodology: M.L., P.J., A.D., T.P., M.C., C.M. Investigation: P.J., A.D., T.P., M.L. Supervision: K.S., C.D., M.L. Writing—original draft: P.J., K.S. Writing—review & editing: P.J., K.S., M.L., C.D.

## Competing interests
The authors declare no competing interests.
