## [Peer Review File · Nature Communications]

Reviewers' Comments:

Reviewer #1:

Remarks to the Author:

The manuscript by Jaitly et al., describes the identification of a CUG clade-restricted gene involved in the cell cycle of these fungal species. The authors undertook a screen for factors that promoted chromosome instability and identified 6 genes altogether, three of which that altered whole chromosome loss. The authors focused on one gene named CSA6 that does not have homologs outside of the CUG clade of fungal species. Overexpression of CSA6 causes cells to arrest during DNA segregation due to spindle checkpoint and loss of CSA6 causes cells to arrest following DNA segregation in mitotic exit. The protein appears to be localized to spindle pole bodies and loss of function mutants can be complemented with CSA6 from the closely-related *C. dubliniensis* species. There is a technical aspect that should probably get a little more attention in the experimental design. In validating the CSA strain and FACS approach, the BFP-GFP+ cells were plated to YPD. In Figure S1F, the legend denotes that only large colonies were assessed for the selected marker combinations although since this is a selection, all colonies should have the same marker composition. Small colonies from plating should also be checked for markers since these appear to constitute most of the recovered cells identified by the designed gates in FACS. This was not done in the previous paper from 2015 cited in the manuscript and would help in understanding the phenotype of these mutants (chromosome loss v. hyperrecombination via chromosome fragmentation).

There are some other aspects to the flow cytometry that could use a bit more understanding. Specifically, cells losing Chr4B are present in the FACS plots. Are these colonies viable, as they wouldn't be expected to be based on the previous inability to recover them from random loss? If not, it would be interesting to know how many strains would be identified based on only including BFP+GFP- cells, which are viable but have potentially undergone CIN. This is the population focused on for all subsequent analysis and knowing of any bias towards loss of one Chr4 homolog or the other would indicate potential alternate functions. It is also somewhat surprising that loss of some Chr4B is not accompanied by some cells showing increased frequency of GFP or BFP signal. Was this seen? If not, what explanation could there be for chromosome loss in some strains but no retention of a trisomy in other cells. The connection between cell cycle arrest observed in the overexpression experiments to the chromosome loss phenotype used to identify strains warrants discussion. WGS of strains overexpressing . This would also provide additional insight into the cell cycle arrest of CSA4-6 since *C. albicans* is usually able to overcome chromosomal imbalance fairly readily.

A large fraction of the cells for mutants CSA4-6 run by flow cytometry and fluorescence microscopy would be expected to be hyphal based on images in Figure S3. The hyphal cells would be expected to give large N content because they would be run as "single cells". A gating strategy that selected for yeast here might explain this but is not present to the best of my knowledge. Also, an explanation of the lack of hyphae in subsequent images would be useful.

CSA6 is first identified by growing the cells and measuring loss of Chr4 by flow cytometry and yet all subsequent work shows that CSA6-OE leads to cell arrest. Growth of the population for flow cytometry suggests that some cells in the population are able to overcome this arrest. Comments on this relationship would be very helpful. It would also be useful to know if these populations contain suppressor mutants that are no longer restricted for growth by CSA6-OE.

Line 300 and Figure S4A – it is not clear that continued overexpression of CSA6 is toxic as the lack of large colony formation in Figure S4A could be caused by continued arrest. Replica plating these dilution sets to plates lacking Dox would demonstrate if cells were arrested or non-viable. This could also be done in liquid culture by washing out the Atc/Dox and assessing cell cycle progression of individuals cells in the population. This also applies to lines 325-326, and the methionine-induced repression experiments. It's not clear this is loss of viability or inability to undergo cell division.

The identity of Csa6 appears to be still fairly nebulous. Mass-spec to identify interaction partners or immuno-precipitation of other spindle pole body subunits would help solidify its position within known cellular structures. The microscopy begins to get to this point but is not conclusive for this level of association around multiple structures in close proximity.

Conservation of CSA6 function across CUG clade species is not compelling based on analysis of only CdCSA6. This can be addressed by either testing a more distantly-related CSA6 ortholog such as from *C. orthopsilosis* complex or these statements regarding conservation across *Candida*

species can be toned down. The expression levels of CdCSA6 compared to CaCSA6 in the ectopic mutants would be useful to know as complementation might occur through differences in function or expression. Overexpression of the CdCSA6 may be sufficient to provide enough functional protein to overcome deficiencies that are not possible in the CaCSA6 complemented strain. The Dcsa6/pMET1-CSA6 strain appears to have very small colonies on plates (i.e., Figure 7D) compared to the WT and the CdCSA6 complemented strain.

More consistent overlap in fluorescence is seen between Csa6 and Spc110 than with Tub4. To help solidify Csa6 as localized to the SPB, it would be helpful to show the localization of Spc110 and Tub4. If they show a similar offset pattern of localization, then it is reasonable to conclude that CSA6 and the two homologs correspond to the spindle pole body.

It is not currently clear how these OE strains are different than previous collections produced by the authors such as <https://www.ncbi.nlm.nih.gov/pmc/articles/PMC3457969/>. The Supplemental Methods also point to previous work in construction of the overexpression collection.

It's not clear why a longer spindle in Figure 5F indicates MT disassembly instead of overextension. This should be clarified.

Line 273-274, the coiled-coil domain is never again mentioned in the manuscript and doesn't seem connected to the rest of the work. This should probably be moved to the supplement.

The images in Figure 5E don't look substantially different from each other with the exception of a single cell in +Met condition in the bottom set. More representative images of the quantification on the right would be helpful here.

Figure 3A can be integrated into other panels or removed as it is redundant with the Supplemental Figure 5E only shows one cell with an extended bud and no aberrant nuclear segregation. An image with more aberrant cell types would be useful here as it would match the distribution shown to the right more accurately than an overwhelming population of segregated but undivided cells.

Line 290: SN148 is not the wildtype background of *C. albicans*. The genetic background should be defined as no wildtype for a species exists.

Use of an induced +Atc negative control strain in Figure S3A would serve as a better control than an uninduced strain that has never seen the induction molecule.

The shades of blue in Fig 2A are too similar to be easily distinguished. A wider color palette would be helpful.

There are a large number of grammatical mistakes throughout the document. This needs careful review for grammar and complete sentence structure.

Some examples are:

Line 77-79: This sentence is a bit hard to follow because of the nested lists and the attachment of chromosome instability at the end. I'd suggest breaking this sentence up.

Line 80: revise as "...associated with the generation of aneuploidy..."

Line 83: "...unicellular and primarily asexual..."

Line 112: "help" should be "helping"

Papers by Burrack L. should be included in descriptions of CENP-A in *C. albicans*.

Line 133-135 seems unnecessary.

Line 175: add "to" before "monitor"

Reviewer #2:

Remarks to the Author:

In the manuscript "A phylogenetically-restricted essential cell cycle progression factor in the human pathogen *Candida albicans*", Jaitly et al. prepared a collection of 1067 inducible overexpression strains and conducted an overexpression screen of ~20% of pathogenic fungus *C. albicans* ORFome. The authors created reporter system on Chr4 and uncovered six genes that control chromosomal stability during cell division. Two out of six genes have been previously implicated with cell cycle. One of six genes with the unknown function is of a special importance, as it encodes a protein, which represents clade-specific cell cycle progression factor that can be served as potential therapeutic target. This study is timely, as only few genes were reported so far to maintain genome stability during cell division. Genes implicated with cell cycle can be drug targets. I recommend this manuscript for publication given that authors address the Comments.

COMMENTS

Several questions have to be addressed in Discussion.

1. Authors have to explain why Chr4 has been chosen for the approach used for the screen.
2. An important question is whether the same set of 6 genes will be found if a different chromosome than Chr4 would be chosen for the reporter system? This question has to be addressed in Discussion.
3. Did authors study chromosome condition outside of Chr4 in FACS-identified cells?

Minor points

The introduction section could be shortened

Reviewer #3:

Remarks to the Author:

In this manuscript, entitled "A phylogenetically-restricted essential cell cycle progression factor..." Jaitly and co-authors from the d'Enfert and Sanyal labs report about their efforts in screening for genes that promote chromosome instability when overexpressed in *Candida albicans*. Within the about 1000 *C. albicans* genes tested, they identify 6 genes that show a reproducible effect upon overexpression. While 5 of these six genes have already been identified (CLB4, ASE1, KIP2, MCM7, BFA1), a sixth one has not been characterized yet and is called CSA6. The protein product of this gene localizes to SPBs throughout the cell cycle. Overexpression of Csa6 arrests the cell with a large bud, replicated DNA and no spindle, due to lack of SPB separation. Deleting the MAD2 gene restores the formation of a spindle and may restore cycling of the cells, suggesting that Csa6 overexpressing cells arrest in the metaphase of mitosis, probably due to spindle assembly defects. In contrast, Csa6 depletion through promoter shutdown causes cells to arrest in late mitosis, with separated SPBs, an elongated spindle, which fails to disassemble, and properly segregated chromosome masses. Remarkably, genes orthologous to CSA6 exist in other CUG-Ser clade yeasts but cannot be traced outside of that group of species. Thus, the CSA6 gene is essential and may offer a useful drug target to treat candidiasis.

Overall, this is an interesting paper both for the biology that it reports and the perspective that it opens for treating fungal infections. The mechanisms of genome plasticity in *Candida albicans* and their role in the remarkable ability of this fungus to adapt and survive to hostile environments and drug treatments have profound implications for our understanding of genome stability and its regulation, evolution and for medicine. The identification of CSA6 has the potential to provide many new insights in all these different areas. Furthermore, the data presented are solid, well documented and convincing. Therefore, I find this paper very valuable. However, addressing at least some of the points below could strengthen this communication substantially further and increase its potential impact.

Main points:

1- The main difficulty of this paper, although one cannot speak of a weakness, is the apparent complexity of the cellular role of Csa6. Whereas Csa6 overexpression seems to arrest the cells at the SAC with an improperly assembled spindle, its depletion leads to a terminal defect later in the cell cycle and arrests the cells in telophase. This is an unusual phenotype in many ways. Addressing the unusual aspects of Csa6 impact on the cell would be very informative and immediately increase the impact of the paper.

First, the overexpression and depletion phenotypes suggest that Csa6 must be tightly controlled to prevent its activation prior to anaphase. However, the protein is constitutively present. If the function of Csa6 is in telophase, why is Csa6 overexpression toxic before that stage? If the role of Csa6 is to promote mitotic exit and hence, Cdk1 inactivation, could it be that excess Csa6 activity early in the cell cycle prevents the proper raise of Cdk1 activity and subsequent mitotic progression? The deletion of MAD2 could help these cells in progressing through the cycle by lowering their need for high Cdk in order to enter anaphase? Monitoring cyclin levels in the wild type and *mad2Δ* cells overexpressing Csa6, compared to endogenous Csa6 levels, could help address this question.

Second, it is very unusual that inactivation of MAD2 restores the ability of defective cells to properly assemble their spindle. This data suggests that the effect of Csa6 overexpression on

metaphase is indirect and caused by the overactivation of the SAC itself in these cells, rather than to a direct effect of Csa6 on spindle assembly. Here again, a premature drop in Cdk1 activity prior to securin degradation could cause premature mitotic exit and prevent proper spindle assembly, explaining the spindle morphology of the Csa6 overexpressing cells.

Third, we lack insights into why Csa6 overexpression causes chromosome mis-segregation. Does the *mad2Δ* mutation restore chromosome stability in these cells, at least to the level of chromosome instability observed in SAC mutant cells, or does it make it even worse? Is Csa6 repression causing chromosome instability as well?

Fourth, the fact that Tem1 localizes to only one of the two SPBs in the Csa6 depleted cells is not very informative. This is also the case in metaphase cells and early anaphase cells. More informative would be to visualize the localization of Cdc15 (which requires tem1 activation to be recruited to SPBs) and to know test whether the different phenotypes of the Csa6 depleted cells (including Tem1 localization) are suppressed upon expression of a catalytically inactive form of the GAP Bub2 (see Scarfone et al., 2015). This would clarify whether Csa6 acts upstream or downstream from Tem1 activation.

Finally, the fact that Csa6 is on the spindle is interesting but knowing on which side of the spindle it localizes would be even more important for allowing clearer interpretations. If it is on the nuclear side, it could give a hint about how it affects chromosome segregation and would suggest that it does not function directly in the MEN. If it is on the cytoplasmic side, it would support a function in the MEN but it would suggest that it does not directly affect spindle assembly and chromosome attachment to the spindle. The data presented speak for Csa6 being on cytoplasmic side but are not addressing this point in a conclusive manner. Investigating how it localizes relative to a central plaque component, such as Spc42 in *S. cerevisiae*, would address that question.

2- It is remarkable that the screen of the authors identifies Kip2 and Bfa1 as well, since Kip2 is involved in spindle orientation and Bfa1 functions in the same pathway as that suggested for Csa6. This begs for more discussion and testing the functional relationships between these genes. For example, does Bfa1 inactivation suppress the deleterious effect of Csa6 depletion? Is Csa6 overexpression still lethal in *bfa1Δ* mutant cells? This could help address the points raised above and give more coherence to the manuscript.

Minor points:

1- The authors do not comment much about it but FACS analysis of the arrested cells indicates that they start to re-replicate their genome as the amount of cells with increased DNA content grows to form a large shoulder on the 4N peak. This observation could support the idea that Cdk1 levels are low in these cells, allowing some over-replication?

2- Are there other genes specific to the CUG-Ser clade the presence of which correlates particularly well with that of Csa6? Are there any changes in Nud1, Bfa1, Bub2 or Spc72 organization that correlate well with the presence of Csa6? This could be very informative about how this new protein inserts itself in the SPB and the MEN pathway.

3- Page 15, line 486: The data suggest that Csa6 affects spindle assembly and not that "Csa6 is required during G2/M for proper assembly of the mitotic spindle". If it were so, Csa6 depletion would cause a spindle assembly defect and these cells would never reach telophase.

Manuscript # NCOMMS-21-40627

Title: A phylogenetically-restricted essential cell cycle progression factor in the human pathogen *Candida albicans*

Authors: Priya Jaitly¹, Mélanie Legrand², Abhijit Das^{1†}, Tejas Patel^{1†}, Murielle Chauvel², Corinne Maufrais^{2,3}, Christophe d'Enfert^{2*} and Kaustuv Sanyal^{1,4*}

Point-by-point responses to reviewers' queries

Reviewer #1

We thank the reviewer for the positive feedback and constructive criticism. The concerns raised by the reviewer are addressed below:

R1.1: There is a technical aspect that should probably get a little more attention in the experimental design. In validating the CSA strain and FACS approach, the BFP-GFP+ cells were plated to YPD. In Figure S1F, the legend denotes that only large colonies were assessed for the selected marker combinations although since this is a selection, all colonies should have the same marker composition. Small colonies from plating should also be checked for markers since these appear to constitute most of the recovered cells identified by the designed gates in FACS. This was not done in the previous paper from 2015 cited in the manuscript and would help in understanding the phenotype of these mutants (chromosome loss v. hyperrecombination via chromosome fragmentation).

Authors: We analyzed small colonies of EV, *CSA1^{CLB4}*, *CSA2^{ASE1}* and *CSA3^{KIP2}* strains for auxotrophic markers. In all these strains, most small colonies (>95%) lost both *HIS1* and *HYG B* while retaining *ARG4*, indicating small colonies are monosomic for Ch4A. This is consistent with our previous work¹. We have included this new data in the revised manuscript (*Fig. S3E and lines 228-230*).

R1.2: There are some other aspects to the flow cytometry that could use a bit more understanding. Specifically, cells losing Chr4B are present in the FACS plots. Are these colonies viable, as they wouldn't be expected to be based on the previous inability to recover them from random loss? If not, it would be interesting to know how many strains would be identified based on only including BFP+GFP- cells, which are viable but have potentially undergone CIN. This is the population focused on for all subsequent analysis and knowing of any bias towards loss of one Chr4 homolog or the other would indicate potential alternate functions.

Au: Perhaps, it is not clear to the reviewer that the GFP/BFP reporter strain that we employed in the current study is different from the one we reported previously by Loll-Krippleber et al, 2015¹. In the current study, we have employed the GFP/BFP strain used by Feri et al, 2016² in which GFP and BFP genes are integrated into the same loci on Ch4A and Ch4B, respectively. Cells losing Ch4B (thus losing BFP and RFP) correspond to BFP⁻GFP⁺ cells that will retain only Ch4A. Depending upon the number of copies of Ch4, cells can either remain monosomic for Ch4A and hence will yield small colonies or become disomic yielding large colonies. We show that BFP⁻GFP⁺ cells are viable in Fig S1F, S3D&E. Cells losing Ch4A (thus losing GFP) correspond to BFP⁺GFP⁻ cells and are inviable².

R1.3: It is also somewhat surprising that loss of some Chr4B is not accompanied by some cells showing increased frequency of GFP or BFP signal. Was this seen? If not, what explanation could there be for chromosome loss in some strains but no retention of a trisomy in other cells.

Au: Indeed, if the chromosome loss event occurs due to a failure in chromosome segregation post-DNA replication, one would expect both – cells that have lost one of the Ch4 homologs as well as cells that have gained an additional Ch4 homolog. While our flow cytometry analysis pipeline allows the detection of cells that have gone from 1x GFP (or BFP) to no GFP (or BFP) signals, detecting cells with an increase in GFP signal – i.e., the transition from 1x GFP (or BFP) to 2x GFP (or BFP) signals by the same method might not be precise. This is because cells gaining an extra copy of Ch4 (Ch4 trisomy) will continue to express both GFP and BFP genes and hence will not be distinguishable from the BFP⁺GFP⁺ cells that are disomic for Ch4.

R1.4: The connection between cell cycle arrest observed in the overexpression experiments to the chromosome loss phenotype used to identify strains warrants discussion. WGS of strains overexpressing. This would also provide additional insight into the cell cycle arrest of CSA4-6 since *C. albicans* is usually able to overcome chromosomal imbalance fairly readily.

Au: To assess the extent of genome changes associated with *CSA6* overexpression, we sequenced the genome of 4 independent large BFP⁻GFP⁺ colonies in which LOH has arisen on Ch4 either spontaneously (in YPD cultures) or as the consequence of *CSA6* overexpression (in YPD + Atc cultures). While we confirmed the LOH events occurring on Ch4, as selected by flow cytometry, we did not observe additional large-scale genome changes other than the LOH event we were selecting for on Ch4. This suggests that overexpression of *CSA6* does not change the nature of the LOH events but rather results in an increased frequency of these events. In budding yeast, depletion of Scm3³ and several *mcm* mutants^{4,5} under restrictive conditions are

shown to cause both cell cycle arrest and chromosome segregation defects. In humans, chromosome mis-segregation or aneuploidy can lead to further genomic instability that ultimately causes cell cycle arrest⁶. Whether cell cycle arrest upon Csa6 overexpression is a cause or a consequence of chromosomal instability remains an enigma. We have incorporated this result and its discussion in the revised manuscript. (Fig. S4 and lines 260-267, 514-518).

R1.5: A large fraction of the cells for mutants CSA4-6 run by flow cytometry and fluorescence microscopy would be expected to be hyphal based on images in Figure S3. The hyphal cells would be expected to give large N content because they would be run as “single cells”. A gating strategy that selected for yeast here might explain this but is not present to the best of my knowledge. Also, an explanation of the lack of hyphae in subsequent images would be useful.

Au: Figure S3A represents the extent of filamentous/polarized growth in *CSA4-6* overexpression mutants after overnight recovery post-8 hours of overexpression by Dox/Atc. In contrast, both the nuclear segregation dynamics and DNA content of *CSA4-6* overexpression mutants were analyzed immediately after 8 hours of overexpression by Dox/Atc without overnight recovery, thus preventing mutants from forming such long filaments. The filamentation in *CSA4-6* mutants is a consequence of continued cell cycle arrest and is distinct from the hyphal state in which cells undergo proper nuclear division⁷. As cells harboring empty vector (EV) do not form such long filaments, we analyzed non-filamentous cells of *CSA4-6* mutants to determine the mitotic stage associated with the 4N shift. We have included the gating strategy used for flow cytometric cell cycle analyses in the revised manuscript (Fig. S13).

R1.6: CSA6 is first identified by growing the cells and measuring loss of Chr4 by flow cytometry and yet all subsequent work shows that CSA6-OE leads to cell arrest. Growth of the population for flow cytometry suggests that some cells in the population are able to overcome this arrest. Comments on this relationship would be very helpful. It would also be useful to know if these populations contain suppressor mutants that are no longer restricted for growth by CSA6-OE.

Au: We agree with the reviewer that some cells in the population were not arrested upon *CSA6* overexpression. We do not know if there is a threshold for *Csa6* overexpression in *C. albicans* which is responsible for causing the mitotic arrest. It is possible that in some cells the level of *Csa6* is below this threshold and that is why they are not arrested. Another possibility is the existence of suppressor mutations. While placing the TAP-tagged copy of *Csa6* under the promoter *P_{TET}*, we serendipitously found two mutations in one of the *E. coli* clones at N421R and P506L positions. Surprisingly, we noticed that cells overexpressing this mutated copy of *Csa6* (*P_{TET}CSA6 N421R P506L-TAP*) do not show reduced growth in presence of Dox. Further analysis involving overexpression of these mutations individually without the epitope tag will validate the existence of suppressor mutations in *Csa6*.

R1.7: Line 300 and Figure S4A – it is not clear that continued overexpression of CSA6 is toxic as the lack of large colony formation in Figure S4A could be caused by continued arrest. Replica plating these dilution sets to plates lacking Dox would demonstrate if cells were arrested or non-viable. This could also be done in liquid culture by washing out the Atc/Dox and assessing cell cycle progression of individuals cells in the population. This also applies to lines 325-326, and the methionine-induced repression experiments. It's not clear this is loss of viability or inability to undergo cell division.

Au: We performed viability assays in the desired strains. Briefly, we grew the strains in presence or absence of either Atc (+/-Atc) or methionine and cysteine (+M+C or -M-C) at various time intervals and spotted them on YPDU agar plates following washing and serial dilutions (materials and methods). We observed that reduced growth upon *Csa6* overexpression is associated with decreased cell viability (Fig. S6B). The cell viability increases markedly when the spindle assembly checkpoint (SAC) was inactivated in the *CSA6^{OE}* strain (Fig. S7B). In contrast, The *CSA6^{PSD}* mutant resumed growth without losing viability when shifted from non-permissive (+M+C) to permissive conditions (-M-C), indicating the arrest phenotype associated with the depletion of *Csa6* is largely reversible (Fig. S8C, D). This is further supported by the increase in *Csa6* levels upon removal of the repressive conditions (Fig. S8E). We have included these new results in the revised manuscript (Fig. S6B, S7B, S8C-E and lines 309-311, 336-337, 373-376).

R1.8: The identity of Csa6 appears to be still fairly nebulous. Mass-spec to identify interaction partners or immuno-precipitation of other spindle pole body subunits would help solidify its position within known cellular structures. The microscopy begins to get to this point but is not conclusive for this level of association around multiple structures in close proximity.

Au: In this study, we identify Csa6 as an essential regulator of cell cycle progression in *C. albicans*. Besides localizing constitutively to the SPBs, Csa6 shows genetic interaction with SPB-associated proteins involved in MEN or SPOC such as Tem1 and Bub2, respectively (Fig. 6G, S10A). The proper mechanism of MEN signaling as well as the structure of SPB is, however, yet to be determined in *C. albicans*. We, therefore, believe that performing mass-spectrometry with either Csa6 or a known SPB component is beyond the scope of this study as we expect this experiment will unravel several SPB-proximal proteins of unknown functions in *C. albicans*.

R1.9: Conservation of CSA6 function across CUG clade species is not compelling based on analysis of only CdCSA6. This can be addressed by either testing a more distantly-related CSA6 ortholog such as from *C. orthopsilosis* complex or these statements regarding conservation across *Candida* species can be toned down. The expression levels of CdCSA6 compared to CaCSA6 in the ectopic mutants would be useful to know as complementation might occur through differences in function or expression. Overexpression of the CdCSA6 may be sufficient to provide enough functional protein to overcome deficiencies that are not possible in the CaCSA6 complemented strain. The $\Delta csa6/p_{MET1}$ -CSA6 strain appears to have very small colonies on plates (i.e., Figure 7D) compared to the WT and the CdCSA6 complemented strain.

Au: We thank the reviewer for this suggestion. To further elucidate the function of Csa6 across CUG clades species, we ectopically expressed GFP-tagged Csa6 of two other *Candida* species, *Candida tropicalis* and *Candida parapsilosis*, in *C. albicans*. The putative Csa6 orthologs are poorly conserved in both *C. tropicalis* (CtCsa6) (Fig. 7D) and *C. parapsilosis* (CpCsa6) (Fig. S12B). Incidentally, CtCsa6, but not CpCsa6, functionally complemented CaCsa6 functions in *C. albicans* (Fig. 7E). Similar to CaCsa6 and CdCsa6, CtCsa6 is also localized to the SPBs throughout the cell cycle in *C. albicans* (Fig. 7F). In contrast, only a small percentage of cells had CpCsa6 localized to the SPBs in *C. albicans* (Fig. S12C). In the majority of cells, CpCsa6 was neither localized to the SPBs nor showed any detectable fluorescence signals (Fig. S12C). This suggests Csa6 is functionally conserved in species closely related to *C. albicans* such as *C. dubliniensis* and *C. tropicalis* but its function might have diverged in more distant species

such as *C. parapsilosis*. The new result and its discussion have been incorporated into the revised manuscript (Fig. 7, S12 and lines 447-457, 558-560).

R1.10: More consistent overlap in fluorescence is seen between Csa6 and Spc110 than with Tub4. To help solidify Csa6 as localized to the SPB, it would be helpful to show the localization of Spc110 and Tub4. If they show a similar offset pattern of localization, then it is reasonable to conclude that CSA6 and the two homologs correspond to the spindle pole body.

Au: We analyzed the localization of Spc110 and Tub4 and observed an offset pattern similar to that of Csa6 and Tub4. In addition, we have also localized homolog of Cmd1, an SPB component belonging to the central plaque in *S. cerevisiae*, with Tub4 and Csa6 in *C. albicans* (also refer to our response to R3.5). Altogether, our results strongly suggest that Csa6 colocalizes with multiple SPB-associated proteins including Tub4, Spc110 and Cmd1 in *C. albicans*. These results have been included in the revised manuscript (*Fig. S5B and lines 281-290*).

R1.11: It is not currently clear how these OE strains are different than previous collections produced by the authors such as <https://www.ncbi.nlm.nih.gov/pmc/articles/PMC3457969/>. The Supplemental Methods also point to previous work in construction of the overexpression collection.

Au: In this study, the overexpression plasmids have been transformed in the *C. albicans* strain CEC5201, a recipient strain that carries the LOH reporter system, which allows us to follow genome stability upon gene expression. In contrast, in the previous study by Chauvel *et al*, 2012⁸, the OE plasmids were transformed in the recipient *C. albicans* wild-type strain CEC161 to identify regulators of morphogenesis and growth rate. In addition, the collection of OE plasmids varies in the two studies: an unbiased collection of 1067 OE plasmids in the current study versus a candidate gene approach using a collection of 384 OE plasmid encoding kinases, phosphatases, transcription factors and other proteins related to signaling used in the study by Chauvel *et al*, 2012⁸.

Minor comments:

R1. 12: It's not clear why a longer spindle in Figure 5F indicates MT disassembly instead of overextension. This should be clarified.

Au: Thank you. We have modified this sentence in the revised manuscript “*indicating a hyper-elongated aberrant mitotic spindle structure in the CSA6^{PSD} mutant*” (*lines 383-384*).

R1.13: Line 273-274, the coiled-coil domain is never again mentioned in the manuscript and doesn't seem connected to the rest of the work. This should probably be moved to the supplement.

Au: We have moved this figure to the supplementary (*Fig. S5A*).

R1.14: The images in Figure 5E don't look substantially different from each other with the exception of a single cell in +Met condition in the bottom set. More representative images of the quantification on the right would be helpful here.

Au: We have added more representative images for Fig. 5E in the supplementary (*Fig. S8B*).

R1.15: Figure 3A can be integrated into other panels or removed as it is redundant with the Supplemental Figure 5E only shows one cell with an extended bud and no aberrant nuclear segregation. An image with more aberrant cell types would be useful here as it would match the distribution shown to the right more accurately than an overwhelming population of segregated but undivided cells.

Au: We apologize for not understanding the former part of this query. Figure 3A illustrates the strategy for overexpressing Csa6 using the tetracycline-inducible promoter and has been shown only once in the manuscript. If the reviewer has a specific comment regarding Figure 3A, we will be happy to modify it. We have added more representative images for Figure 5E in the supplementary as mentioned above in R1.14.

R1.16: Line 290: SN148 is not the wildtype background of *C. albicans*. The genetic background should be defined as no wildtype for a species exists.

Au: We have incorporated the suggested modification in the revised manuscript “*For this, we again made use of the inducible P_{TET} promoter system to generate a $CSA6^{OE}$ strain (CaPJ176, $P_{TET}CSA6$) in the SN148 genetic background of *C. albicans*” (lines 298-300).*

R1.17: Use of an induced +Atc negative control strain in Figure S3A would serve as a better control than an uninduced strain that has never seen the induction molecule.

Au: Thank you. We have replaced the uninduced strain in Fig. S3A with the EV strain treated with Atc.

R1.18: The shades of blue in Fig 2A are too similar to be easily distinguished. A wider color palette would be helpful.

Au: Thank you. We have changed the color scheme for Fig 2A.

R1.19: There are a large number of grammatical mistakes throughout the document. This needs careful review for grammar and complete sentence structure.

Au: Thank you. We have incorporated the below-mentioned changes in the revised manuscript.

Some examples are:

Line 77-79: This sentence is a bit hard to follow because of the nested lists and the attachment of chromosome instability at the end. I'd suggest breaking this sentence up.

As suggested, we have broken this sentence into two “*Genome instability can occur as a consequence of either point mutations, insertions and deletions of bases in specific genes or gain, loss and rearrangements of chromosomes. The latter is also referred to as chromosome instability (CIN)*” (lines 74-77).

Line 80: revise as “...associated with the generation of aneuploidy...”

Thank you. We have revised this sentence (lines 77-78).

Line 83: “...unicellular and primarily asexual..”

Thank you. We have incorporated this change (lines 80-81).

Line 112: “help” should be “helping”

Thank you. We have changed this (line 108).

Papers by Burrack L. should be included in descriptions of CENP-A in *C. albicans*.

We have included this citation in the revised manuscript (line 119).

Line 133-135 seems unnecessary.

We have removed this sentence in the revised version.

Line 175: add “to” before “monitor”

Thank you. We have corrected the mistake in the revised manuscript (line 167).

Reviewer#2

The reviewer notes that “*this study is timely, as only few genes were reported so far to maintain genome stability during cell division*”. We thank the reviewer for the encouraging comments. The reviewer also raised concerns that we have addressed below:

COMMENTS

Several questions have to be addressed in Discussion.

R2-1: Authors have to explain why Chr4 has been chosen for the approach used for the screen.

Au: This has already been described in¹. The LOH reporter system relies on the construction of an artificial locus on Ch4 in the intergenic region between the *PGA59* and *PGA62* open reading frames (ORFs). The d'Enfert Lab has previously shown that the 9170 bp region, that separates the start codons of *PGA59* and *PGA62*, contains a single 309 bp putative ORF, *orf19.2766*, that shows poor coding probability according to a GenMark analysis, and is absent from the genome of *C. albicans* strain WO-1 as well as other *Candida* species such as *C. lusitaniae*, *C. tropicalis* and *C. guilliermondii*. Consistently, no expression of *orf19.2766* was detected by quantitative RT-PCR analysis in YPD or Lee's medium, while *PGA59* and *PGA62* were expressed in these conditions (data not shown)⁹. This 9-kb region has also been used previously as a platform for the integration of *C. albicans* two-hybrid plasmids¹⁰, and its modification is thought to be neutral to *C. albicans* biology, as its deletion did not result in any phenotype⁹.

We agree with the reviewer that the explanation why Chr4 was chosen should be easily accessible. Therefore, rather than presenting in details our choice of Chr4 (which has been previously presented in details in Loll-Krippelber et al, 2015¹, we modified lines 151 to 153 “*In the LOH reporter system, the GFP and BFP genes (associated with the ARG4 and HIS1 auxotrophic markers, respectively) have been integrated at the identical intergenic locus on the left arm of both homologs of chromosome 4, Ch4A and Ch4B, respectively*”¹¹.

R2-2: An important question is whether the same set of 6 genes will be found if a different chromosome than Chr4 would be chosen for the reporter system? This question has to be addressed in Discussion.

Au: The six *CSA* genes (including *CSA6*) identified in the study are important for mitotic progression in *C. albicans* or related yeast species. In fact, two *CSA* genes, *CSA2^{ASE1}* and *CSA5^{BFA1}* have also been identified in the previous screens for CIN in *S. cerevisiae*^{11, 12}. Therefore, we believe that overexpression of these six genes will induce CIN in *C. albicans*, regardless of the choice of the chromosome under investigation. However, as chromosome-specific CIN events have also been reported in *C. albicans*^{13, 14}, there is a possibility of finding genes that specifically cause Ch4 instability using our GFP/BFP-based reporter system. We have included this point in the discussion of the revised manuscript (*lines 471-473*).

R2-3: Did authors study chromosome condition outside of Chr4 in FACS-identified cells?

Au: This is also addressed in R1.4. Briefly, we performed whole-genome sequencing of 4 independent large BFP⁻GFP⁺ colonies in which LOH has arisen on Ch4 either spontaneously (in YPD cultures) or as the consequence of *CSA6* overexpression (in YPD + Atc cultures). We did not detect any additional large-scale genome changes other than the CIN event we selected for Ch4 (the loss of the *BFP-HIS1* marker). (*Lines 260-267*).

Minor points

R2-4: The introduction section could be shortened.

Au: We have shortened the introduction in the revised manuscript.

Reviewer#3

The reviewer acknowledges “*this is an interesting paper both for the biology that it reports and the perspective that it opens for treating fungal infections*”. We thank the reviewer for the encouraging comments and valuable suggestions. The reviewer also raised concerns that we have addressed below:

The main difficulty of this paper, although one cannot speak of a weakness, is the apparent complexity of the cellular role of Csa6. Whereas Csa6 overexpression seems to arrest the cells at the SAC with an improperly assembled spindle, its depletion leads to a terminal defect later in the cell cycle and arrests the cells in telophase. This is an unusual phenotype in many ways. Addressing the unusual aspects of Csa6 impact on the cell would be very informative and immediately increase the impact of the paper.

We thank the reviewer for pointing out the challenges associated with understanding the biological functions of Csa6. We have incorporated these points in the discussion of our revised manuscript (*lines 464-467*).

First, the overexpression and depletion phenotypes suggest that Csa6 must be tightly controlled to prevent its activation prior to anaphase. However, the protein is constitutively present. If the function of Csa6 is in telophase, why is Csa6 overexpression toxic before that stage? If the role of Csa6 is to promote mitotic exit and hence, Cdk1 inactivation, could it be that excess Csa6 activity early in the cell cycle prevents the proper raise of Cdk1 activity and subsequent mitotic progression? The deletion of MAD2 could help these cells in progressing through the cycle by lowering their need for high Cdk in order to enter anaphase? Monitoring cyclin levels in the wild type and *mad2Δ* cells overexpressing Csa6, compared to endogenous Csa6 levels, could help address this question.

Au: We thank the reviewer for proposing these excellent suggestions. In *S. cerevisiae*, four B-type mitotic cyclins—Clb1 to Clb4, with Clb2 as the major mitotic cyclin¹⁵, promote transition from G2 to M phase. Strikingly, a *P_{GAL10}CLB2 clb1 clb2 clb3 clb4* strain in *S. cerevisiae* arrests at the G2/M stage under repressive conditions with unsegregated DNA masses and SPBs in a side-by-side configuration¹⁶. In contrast, Clb2 depleted cells in *C. albicans* arrest at the late anaphase/telophase stage with 4N equivalent DNA content¹⁷. This suggests a potential rewiring in cyclin-dependent regulation of Cdk1 in *C. albicans*.

To test if Csa6 overexpression leads to a concomitant reduction in levels of cyclins resulting in Cdk1 inactivation at an earlier cell cycle stage, we performed additional experiments to monitor Clb2 levels in the *CSA6^{OE}* mutant. Due to the technical difficulties associated with

synchronizing *C. albicans* cells, we decided to measure the Clb2 levels either at the early S (hydroxyurea-arrested) or at the G2/M (nocodazole-arrested) stage in cells carrying EV. These levels were then compared with the Clb2 levels of the *CSA6^{OE}* strain arrested at the G2/M stage following Atc treatment (Fig. 3D, S11A, B). While we confirmed lower levels of Clb2 in the early S-phase as reported previously¹⁷, we found no significant difference in Clb2 levels between G2/M-arrested cells of EV and Atc-treated cells of the *CSA6^{OE}* strain (Fig. S11A, B). These results suggest that Csa6 overexpression does not cause any major alterations in the mitotic cyclin Clb2 levels at the G2/M stage of the cell cycle. There is also a possibility of alternate mechanisms, other than cyclin degradation, leading to Cdk1 inactivation in the *CSA6^{OE}* mutant. We have included this result and its discussion in the revised manuscript (Fig. S11 and lines 415-425, 544-548).

R3.2: Second, it is very unusual that inactivation of MAD2 restores the ability of defective cells to properly assemble their spindle. This data suggests that the effect of Csa6 overexpression on metaphase is indirect and caused by the overactivation of the SAC itself in these cells, rather than to a direct effect of Csa6 on spindle assembly. Here again, a premature drop in Cdk1 activity prior to securin degradation could cause premature mitotic exit and prevent proper spindle assembly, explaining the spindle morphology of the Csa6 overexpressing cells.

Au: This has been addressed above in R3.1.

R3.3: Third, we lack insights into why Csa6 overexpression causes chromosome mis-segregation. Does the *mad2Δ* mutation restore chromosome stability in these cells, at least to the level of chromosome instability observed in SAC mutant cells, or does it make it even worse? Is Csa6 repression causing chromosome instability as well?

Au: This is also addressed in R1.4. In budding yeast, depletion of Scm3³ and several *mcm* mutants^{4, 5} under restrictive conditions are shown to cause both cell cycle arrest and chromosome segregation defects. In humans, chromosome mis-segregation or aneuploidy can lead to further genomic instability that ultimately causes cell cycle arrest⁶. Whether cell cycle arrest upon Csa6 overexpression is a cause or a consequence of chromosomal instability remains an enigma. Unfortunately, due to the conflict of selectable markers and other technical difficulties, we could not generate the desired *C. albicans* strains to determine CIN frequency in *mad2* or *CSA6^{PSD}* genetic background.

R3.4: Fourth, the fact that Tem1 localizes to only one of the two SPBs in the Csa6 depleted cells is not very informative. This is also the case in metaphase cells and early anaphase cells. More informative would be to visualize the localization of Cdc15 (which requires tem1 activation to be recruited to SPBs) and to know test whether the different phenotypes of the Csa6 depleted cells (including Tem1 localization) are suppressed upon expression of a catalytically inactive form of the GAP Bub2 (see Scarfone et al., 2015). This would clarify whether Csa6 acts upstream or downstream from Tem1 activation.

Au: We thank the reviewer for suggesting this experiment. In *S. cerevisiae*, a two-component GTPase-activating protein (GAP) complex consisting of Bub2 and Bfa1 is known to prevent mitotic exit by stimulating Tem1 GTPase activity¹⁸. To determine if Csa6 functions further upstream of Tem1, we sought to delete both copies of Bub2 in *CSA6*^{PSD} mutant as Bub2 but not Bfa1 carries a conserved GAP domain^{19, 20}. Strikingly, we observed a better growth of *CSA6*^{PSD} mutant under non-permissive conditions in the absence of Bub2 (Fig. S10A). On the other hand, the growth defects of the Tem1 conditional mutant in *C. albicans*²¹ were not rescued upon Bub2 deletion (Fig. S10B). This is somewhat comparable to the deletion of Bub2 in *S. cerevisiae* that was shown to suppress the growth defect of a mitotic exit mutant *cdc5-1*²² but not the *tem1-3* conditional mutant²⁰. These results suggest that Csa6 acts upstream of Tem1 in *C. albicans*. We have included this new result in the revised manuscript (Fig. S10 and lines 402-413).

We also tried to epitope tag the C-term of Cdc15 with GFP in *C. albicans*. While we could localize Cdc15 to the SPBs as reported previously²³, the poor localization of Cdc15 made the overall analysis difficult and inconclusive.

R3.5: Finally, the fact that Csa6 is on the spindle is interesting but knowing on which side of the spindle it localizes would be even more important for allowing clearer interpretations. If it is on the nuclear side, it could give a hint about how it affects chromosome segregation and would suggest that it does not function directly in the MEN. If it is on the cytoplasmic side, it would support a function in the MEN but it would suggest that it does not directly affect spindle assembly and chromosome attachment to the spindle. The data presented speak for Csa6 being

on cytoplasmic side but are not addressing this point in a conclusive manner. Investigating how it localizes relative to a central plaque component, such as Spc42 in *S. cerevisiae*, would address that question.

Au: In *S. cerevisiae*, five SPB proteins, namely Spc42, Spc29, Cnm67 (C-terminus), Spc110 (C-terminus) and Cmd1 have been associated with central plaque^{24, 25}. While we could find a homolog of Cmd1 (orf 19.4413) and Spc110 (orf 19.3100) in *C. albicans*, the presence of the remaining three SPB proteins remains elusive in this ascomycete. Therefore, we checked Cmd1 localization with Tub4 and Csa6. Similar to ScCmd1, CaCmd1 also localizes to the spindle pole bodies throughout the cell cycle²⁶. In addition, we observed ring-like fluorescence signals of Cmd1 near the bud neck and cables of fluorescence in the buds²⁶. Although we found Csa6 in close proximity to Cmd1, we do not know to which compartment of SPBs these two proteins belong as the ultrastructure of SPBs in *C. albicans* remains unknown. Nevertheless, our results strongly suggest that Csa6 is constitutively localized to the SPBs in *C. albicans*. We have included this data in the revised manuscript (*Fig S5C, 2E and lines 284-290*).

R3.6: It is remarkable that the screen of the authors identifies Kip2 and Bfa1 as well, since Kip2 is involved in spindle orientation and Bfa1 functions in the same pathway as that suggested for Csa6. This begs for more discussion and testing the functional relationships between these genes. For example, does Bfa1 inactivation suppress the deleterious effect of Csa6 depletion? Is Csa6 overexpression still lethal in *bfa1Δ* mutant cells? This could help address the points raised above and give more coherence to the manuscript.

Au: In *S. cerevisiae*, Bfa1-Bub2 forms a two-component GAP complex to negatively regulate the Tem1 GTPase. Strikingly, deletion of either Bfa1 or Bub2 suppresses the growth defects associated with the *cdc5-1* mutation, which is defective in mitotic exit at restrictive temperature²². We found Bub2 deletion rescues the growth defect associated with Csa6

depletion (Fig S10). A slight growth improvement, if any, was observed upon Bub2 deletion in *CSA6^{OE}* strain. Altogether our results suggest that Csa6 functions at the proximal end of MEN signaling. These results have been included in the revised manuscript (Fig. S7E, F, S10 and lines 353-355, 402-413).

Minor points:

1- The authors do not comment much about it but FACS analysis of the arrested cells indicates that they start to re-replicate their genome as the amount of cells with increased DNA content grows to form a large shoulder on the 4N peak. This observation could support the idea that Cdk1 levels are low in these cells, allowing some over-replication?

Au: This has been addressed in R3.1.

2- Are there other genes specific to the CUG-Ser clade the presence of which correlates particularly well with that of Csa6? Are there any changes in Nud1, Bfa1, Bub2 or Spc72 organization that correlate well with the presence of Csa6? This could be very informative about how this new protein inserts itself in the SPB and the MEN pathway.

Au: We are not aware of any such protein whose presence correlates with that of Csa6. We could find putative homolog of Nud1 (orf 19.6789), Bfa1 (orf 19.6080), Bub2 (orf 19.5827) and Spc72 (orf 19.6583) in *C. albicans*. Except for Bub2, the remaining proteins have not been functionally verified in *C. albicans*.

3- Page 15, line 486: The data suggest that Csa6 affects spindle assembly and not that “Csa6 is required during G2/M for proper assembly of the mitotic spindle”. If it were so, Csa6 depletion would cause a spindle assembly defect and these cells would never reach telophase.

Au: We thank the reviewer for pointing this out. We have removed this statement from the discussion.

References

1. Loll-Krippelber R, *et al.* A FACS-optimized screen identifies regulators of genome stability in *Candida albicans*. *Eukaryot Cell* **14**, 311-322 (2015).
2. Feri A, *et al.* Analysis of Repair Mechanisms following an Induced Double-Strand Break Uncovers Recessive Deleterious Alleles in the *Candida albicans* Diploid Genome. *mBio* **7**, (2016).

3. Mizuguchi G, Xiao H, Wisniewski J, Smith MM, Wu C. Nonhistone Scm3 and histones CenH3-H4 assemble the core of centromere-specific nucleosomes. *Cell* **129**, 1153-1164 (2007).
4. Yan H, Gibson S, Tye BK. Mcm2 and Mcm3, two proteins important for ARS activity, are related in structure and function. *Genes Dev* **5**, 944-957 (1991).
5. Elble R, Tye BK. Chromosome loss, hyperrecombination, and cell cycle arrest in a yeast mcm1 mutant. *Mol Biol Cell* **3**, 971-980 (1992).
6. Santaguida S, *et al.* Chromosome Mis-segregation Generates Cell-Cycle-Arrested Cells with Complex Karyotypes that Are Eliminated by the Immune System. *Dev Cell* **41**, 638-651 e635 (2017).
7. Finley KR, Berman J. Microtubules in *Candida albicans* hyphae drive nuclear dynamics and connect cell cycle progression to morphogenesis. *Eukaryot Cell* **4**, 1697-1711 (2005).
8. Chauvel M, *et al.* A versatile overexpression strategy in the pathogenic yeast *Candida albicans*: identification of regulators of morphogenesis and fitness. *PLoS One* **7**, e45912 (2012).
9. Moreno-Ruiz E, *et al.* The GPI-modified proteins Pga59 and Pga62 of *Candida albicans* are required for cell wall integrity. *Microbiology (Reading)* **155**, 2004-2020 (2009).
10. Stynen B, Van Dijck P, Tournu H. A CUG codon adapted two-hybrid system for the pathogenic fungus *Candida albicans*. *Nucleic Acids Res* **38**, e184 (2010).
11. Stevenson LF, Kennedy BK, Harlow E. A large-scale overexpression screen in *Saccharomyces cerevisiae* identifies previously uncharacterized cell cycle genes. *Proc Natl Acad Sci U S A* **98**, 3946-3951 (2001).
12. Duffy S, *et al.* Overexpression screens identify conserved dosage chromosome instability genes in yeast and human cancer. *Proc Natl Acad Sci U S A* **113**, 9967-9976 (2016).
13. Kravets A, Yang F, Bethlenny G, Cao Y, Sherman F, Rustchenko E. Adaptation of *Candida albicans* to growth on sorbose via monosomy of chromosome 5 accompanied by duplication of another chromosome carrying a gene responsible for sorbose utilization. *FEMS Yeast Res* **14**, 708-713 (2014).
14. Miller MG, Johnson AD. White-opaque switching in *Candida albicans* is controlled by mating-type locus homeodomain proteins and allows efficient mating. *Cell* **110**, 293-302 (2002).
15. Richardson H, Lew DJ, Henze M, Sugimoto K, Reed SI. Cyclin-B homologs in *Saccharomyces cerevisiae* function in S phase and in G2. *Genes Dev* **6**, 2021-2034 (1992).

16. Fitch I, *et al.* Characterization of four B-type cyclin genes of the budding yeast *Saccharomyces cerevisiae*. *Mol Biol Cell* **3**, 805-818 (1992).
17. Bensen ES, Clemente-Blanco A, Finley KR, Correa-Bordes J, Berman J. The mitotic cyclins Clb2p and Clb4p affect morphogenesis in *Candida albicans*. *Mol Biol Cell* **16**, 3387-3400 (2005).
18. Hotz M, Barral Y. The Mitotic Exit Network: new turns on old pathways. *Trends Cell Biol* **24**, 145-152 (2014).
19. Neuwald AF. A shared domain between a spindle assembly checkpoint protein and Ypt/Rab-specific GTPase-activators. *Trends Biochem Sci* **22**, 243-244 (1997).
20. Scarfone I, Venturetti M, Hotz M, Lengefeld J, Barral Y, Piatti S. Asymmetry of the budding yeast Tem1 GTPase at spindle poles is required for spindle positioning but not for mitotic exit. *PLoS Genet* **11**, e1004938 (2015).
21. Milne SW, *et al.* Role of *Candida albicans* Tem1 in mitotic exit and cytokinesis. *Fungal Genet Biol* **69**, 84-95 (2014).
22. Ro HS, Song S, Lee KS. Bfa1 can regulate Tem1 function independently of Bub2 in the mitotic exit network of *Saccharomyces cerevisiae*. *Proc Natl Acad Sci U S A* **99**, 5436-5441 (2002).
23. Bates S. *Candida albicans* Cdc15 is essential for mitotic exit and cytokinesis. *Sci Rep* **8**, 8899 (2018).
24. Klenchin VA, Frye JJ, Jones MH, Winey M, Rayment I. Structure-function analysis of the C-terminal domain of CNM67, a core component of the *Saccharomyces cerevisiae* spindle pole body. *J Biol Chem* **286**, 18240-18250 (2011).
25. Cavanaugh AM, Jaspersen SL. Big Lessons from Little Yeast: Budding and Fission Yeast Centrosome Structure, Duplication, and Function. *Annu Rev Genet* **51**, 361-383 (2017).
26. Moser MJ, Flory MR, Davis TN. Calmodulin localizes to the spindle pole body of *Schizosaccharomyces pombe* and performs an essential function in chromosome segregation. *J Cell Sci* **110 (Pt 15)**, 1805-1812 (1997).

Reviewers' Comments:

Reviewer #1:

Remarks to the Author:

I greatly appreciate the effort by the authors to address all concerns. All major concerns have been addressed, and the manuscript is a very nice, comprehensive work.

Two quick notes:

1. The "," on line 115 should be removed.
2. Chr2 has undergone LOH in all sequenced progeny in Figure S4 compared to the SC5314 control. While unlikely to be the result of selection by FACS, this might deserve a quick comment.

Reviewer #3:

Remarks to the Author:

The authors have added substantial information to their manuscript and although not all points could be addressed and several aspects of Csa6 function remain enigmatic, this reviewer feels that the amount and novelty of the information provided by this manuscript makes it a very useful paper and an excellent candidate for publication in Nature Communication.

Point-by-point responses to reviewers' queries (second round of revision)

Reviewer #1

Remarks to the Author:

I greatly appreciate the effort by the authors to address all concerns. All major concerns have been addressed, and the manuscript is a very nice, comprehensive work.

Au: We thank the reviewer for their response.

Two quick notes:

1. The "," on line 115 should be removed.

Au: Thank you. We have incorporated the suggested modification.

2. Chr2 has undergone LOH in all sequenced progeny in Figure S4 compared to the SC5314 control. While unlikely to be the result of selection by FACS, this might deserve a quick comment.

Au: Additional LOH events, such as LOH on Chr2, have previously been described in our parental strain that carries the BFP/GFP system¹. We have included this point in the legend of figure S4.

Reviewer #3

Remarks to the Author:

The authors have added substantial information to their manuscript and although not all points could be addressed and several aspects of Csa6 function remain enigmatic, this reviewer feels that the amount and novelty of the information provided by this manuscript makes it a very useful paper and an excellent candidate for publication in Nature Communication.

Au: We thank the reviewer for their recommendation.

References

1. Loll-Krippelber R, *et al.* A FACS-optimized screen identifies regulators of genome stability in *Candida albicans*. *Eukaryot Cell* **14**, 311-322 (2015).
2. Feri A, *et al.* Analysis of Repair Mechanisms following an Induced Double-Strand Break Uncovers Recessive Deleterious Alleles in the *Candida albicans* Diploid Genome. *mBio* **7**, (2016).
3. Mizuguchi G, Xiao H, Wisniewski J, Smith MM, Wu C. Nonhistone Scm3 and histones CenH3-H4 assemble the core of centromere-specific nucleosomes. *Cell* **129**, 1153-1164 (2007).
4. Yan H, Gibson S, Tye BK. Mcm2 and Mcm3, two proteins important for ARS activity, are related in structure and function. *Genes Dev* **5**, 944-957 (1991).
5. Elble R, Tye BK. Chromosome loss, hyperrecombination, and cell cycle arrest in a yeast *mcm1* mutant. *Mol Biol Cell* **3**, 971-980 (1992).
6. Santaguida S, *et al.* Chromosome Mis-segregation Generates Cell-Cycle-Arrested Cells with Complex Karyotypes that Are Eliminated by the Immune System. *Dev Cell* **41**, 638-651 e635 (2017).
7. Finley KR, Berman J. Microtubules in *Candida albicans* hyphae drive nuclear dynamics and connect cell cycle progression to morphogenesis. *Eukaryot Cell* **4**, 1697-1711 (2005).
8. Chauvel M, *et al.* A versatile overexpression strategy in the pathogenic yeast *Candida albicans*: identification of regulators of morphogenesis and fitness. *PLoS One* **7**, e45912 (2012).
9. Moreno-Ruiz E, *et al.* The GPI-modified proteins Pga59 and Pga62 of *Candida albicans* are required for cell wall integrity. *Microbiology (Reading)* **155**, 2004-2020 (2009).
10. Stynen B, Van Dijck P, Tournu H. A CUG codon adapted two-hybrid system for the pathogenic fungus *Candida albicans*. *Nucleic Acids Res* **38**, e184 (2010).
11. Stevenson LF, Kennedy BK, Harlow E. A large-scale overexpression screen in *Saccharomyces cerevisiae* identifies previously uncharacterized cell cycle genes. *Proc Natl Acad Sci U S A* **98**, 3946-3951 (2001).
12. Duffy S, *et al.* Overexpression screens identify conserved dosage chromosome instability genes in yeast and human cancer. *Proc Natl Acad Sci U S A* **113**, 9967-9976 (2016).
13. Kravets A, Yang F, Bethlendy G, Cao Y, Sherman F, Rustchenko E. Adaptation of *Candida albicans* to growth on sorbose via monosomy of chromosome 5 accompanied by duplication of another chromosome carrying a gene responsible for sorbose utilization. *FEMS Yeast Res* **14**, 708-713 (2014).

14. Miller MG, Johnson AD. White-opaque switching in *Candida albicans* is controlled by mating-type locus homeodomain proteins and allows efficient mating. *Cell* **110**, 293-302 (2002).
15. Richardson H, Lew DJ, Henze M, Sugimoto K, Reed SI. Cyclin-B homologs in *Saccharomyces cerevisiae* function in S phase and in G2. *Genes Dev* **6**, 2021-2034 (1992).
16. Fitch I, *et al.* Characterization of four B-type cyclin genes of the budding yeast *Saccharomyces cerevisiae*. *Mol Biol Cell* **3**, 805-818 (1992).
17. Bensen ES, Clemente-Blanco A, Finley KR, Correa-Bordes J, Berman J. The mitotic cyclins Clb2p and Clb4p affect morphogenesis in *Candida albicans*. *Mol Biol Cell* **16**, 3387-3400 (2005).
18. Hotz M, Barral Y. The Mitotic Exit Network: new turns on old pathways. *Trends Cell Biol* **24**, 145-152 (2014).
19. Neuwald AF. A shared domain between a spindle assembly checkpoint protein and Ypt/Rab-specific GTPase-activators. *Trends Biochem Sci* **22**, 243-244 (1997).
20. Scarfone I, Venturetti M, Hotz M, Lengefeld J, Barral Y, Piatti S. Asymmetry of the budding yeast Tem1 GTPase at spindle poles is required for spindle positioning but not for mitotic exit. *PLoS Genet* **11**, e1004938 (2015).
21. Milne SW, *et al.* Role of *Candida albicans* Tem1 in mitotic exit and cytokinesis. *Fungal Genet Biol* **69**, 84-95 (2014).
22. Ro HS, Song S, Lee KS. Bfa1 can regulate Tem1 function independently of Bub2 in the mitotic exit network of *Saccharomyces cerevisiae*. *Proc Natl Acad Sci U S A* **99**, 5436-5441 (2002).
23. Bates S. *Candida albicans* Cdc15 is essential for mitotic exit and cytokinesis. *Sci Rep* **8**, 8899 (2018).
24. Klenchin VA, Frye JJ, Jones MH, Winey M, Rayment I. Structure-function analysis of the C-terminal domain of CNM67, a core component of the *Saccharomyces cerevisiae* spindle pole body. *J Biol Chem* **286**, 18240-18250 (2011).
25. Cavanaugh AM, Jaspersen SL. Big Lessons from Little Yeast: Budding and Fission Yeast Centrosome Structure, Duplication, and Function. *Annu Rev Genet* **51**, 361-383 (2017).
26. Moser MJ, Flory MR, Davis TN. Calmodulin localizes to the spindle pole body of *Schizosaccharomyces pombe* and performs an essential function in chromosome segregation. *J Cell Sci* **110** (Pt 15), 1805-1812 (1997).